# Independently evolved viral effectors convergently suppress DELLA protein SLR1-mediated broad-spectrum antiviral immunity in rice

Lulu Li[1,2], Hehong Zhang[2], Zihang Yang[2], Chen Wang[1,2], Shanshan Li[1,2], Chen Cao[2], Tongsong Yao[1,2], Zhongyan Wei[2], Yanjun Li[2], Jianping Chen ⓘ [1,2] ✉ & Zongtao Sun ⓘ [2] ✉

Plant viruses adopt diverse virulence strategies to inhibit host antiviral defense. However, general antiviral defense directly targeted by different types of plant viruses have rarely been studied. Here, we show that the single rice DELLA protein, SLENDER RICE 1 (SLR1), a master negative regulator in Gibberellin (GA) signaling pathway, is targeted by several different viral effectors for facilitating viral infection. Viral proteins encoded by different types of rice viruses all directly trigger the rapid degradation of SLR1 by promoting association with the GA receptor OsGID1. SLR1-mediated broad-spectrum resistance was subverted by these independently evolved viral proteins, which all interrupted the functional crosstalk between SLR1 and jasmonic acid (JA) signaling. This decline of JA antiviral further created the advantage of viral infection. Our study reveals a common viral counter-defense strategy in which different types of viruses convergently target SLR1-mediated broad-spectrum resistance to benefit viral infection in the monocotyledonous crop rice.

Plants are frequently challenged by a range of viral pathogens in both natural and agricultural ecosystems. Successful infection by viruses with diverse genome sequences and virion structure leads to enormous losses in crop yields[1,2]. In the arms race between viruses and their hosts, very diverse plant viruses have developed some strategies that target common antiviral pathways. For example, most plant viruses interfere with RNA silencing-mediated antiviral defense by encoding various type of RNA silencing suppressors[3–5]. Understanding such conserved pathogenic mechanisms is vital for understanding viral pathogenicity and developing broad-spectrum antiviral strategies.

Among the various RNA viruses infecting rice, *Rice black-streaked dwarf virus* (RBSDV), *Southern rice black-streaked dwarf virus* (SRBSDV), *Rice stripe virus* (RSV) and the newly emerged *Rice stripe mosaic virus* (RSMV) are perhaps the most successful as they cause

serious threats to stable crop yields. RBSDV and SRBSDV are closely-related members of the genus *Fijivirus* (family *Reoviridae*) and can cause similar dwarfing symptoms in rice[6]. Their genome consists of ten segments of double-stranded RNA that encode a total of thirteen proteins[7,8]. Not all of the viral proteins have been clearly identified but P8 and P10 have been generally recognized as the core viral capsid and outer capsid proteins respectively[9,10]. RSV and RSMV have single-stranded genomes and are classified in the genera *Tenuivirus* (family *Phenuiviridae*) and *Cytorhabdovirus* (family *Rhabdoviridae*), respectively. The genome of RSV comprises four RNAs that encode a total of seven proteins by an ambisense coding strategy, and the P2 protein has been identified as an RNA silencing suppressor[11–13]. RSV-infected plants typically display chlorosis, weakness and necrosis in emerging leaves, and as a result their growth is stunted. RSMV is the only

[1]College of Plant Protection, Nanjing Agricultural University, Nanjing 210095, China. [2]State Key Laboratory for Managing Biotic and Chemical Threats to the Quality and Safety of Agro-products, Key Laboratory of Biotechnology in Plant Protection of MOA of China and Zhejiang Province, Institute of Plant Virology, Ningbo University, Ningbo 315211, China. ✉e-mail: jianpingchen@nbu.edu.cn; sunzongtao@nbu.edu.cn

cytorhabdovirus so far reported from naturally-infected rice plants. The genomes of rhabdoviruses encode a minimum of five canonical proteins in the following conserved order: nucleocapsid protein (N), phosphoprotein (P), matrix protein (M), glycoprotein (G) and large polymerase protein (L) (3′-N-P-M-G-L-5′). RSMV-infected plants are slightly dwarfed and have twisted leaves with yellow stripes and mosaicism[14,15]. We recently reported that viral proteins from these very diverse plant RNA viruses (SRBSDV SP8, RBSDV P8, RSV P2, and RSMV M) all interacted with the same targets (auxin response transcription factor OsARF17, JA signaling central components OsJAZ and OsMYC2/3) to repress jasmonate (JA) and auxin signaling, making plants more susceptible to infection by their respective viruses[16,17]. Whether these distinct viral proteins interact with any other common host factor(s) remains to be determined.

Gibberellin (GA), one of the tetracyclic diterpenoid plant hormones, plays essential roles in plant growth and development[18,19]. As master growth repressors, DELLA proteins (SLR1 in rice) are key components of GA signaling. DELLA proteins generally consist of an N-terminal DELLA/TVHYNP motif and a C-terminal GRAS domain. The GA signal is perceived by its receptor GIBBERELLIN INSENSITIVE DWARF1 (GID1) that undergoes a conformational change and then promotes the formation of the GA-GID1-DELLA complex, which is subsequently ubiquitinated by the Skp1-Cullin-F-box (SCF) E3 ubiquitin-ligase complex via the F-box protein (SLEEPY1 [SLY1] in Arabidopsis and GIBBERELLIN INSENSITIVE DWARF2 [GID2] in rice)[20,21], and then degraded by the 26S proteasome, resulting in the downstream release of the DELLA-repressed GA responses as plants grow and develop[22]. Recent evidence has greatly expanded our understanding of GA signaling, especially the roles of DELLA hubs in coordinating diverse processes throughout plant growth and development. For instance, DELLAs can physically combine with the bHLH transcription factor PIFs, thus inducing PIF degradation, which contributes to coordination of light signals during growth and development in Arabidopsis[23]. In addition to this well-established model in which plant growth is fine-tuned to adapt to changing environmental conditions, GA and its DELLA hubs are also widely involved in plant responses to attack by pathogenic fungi or bacteria[24–26]. DELLA proteins boost basal Arabidopsis immunity against necrotrophs by positively integrating the JA signaling pathway, while DELLA protein SLR1 enhances rice defenses against the hemibiotrophic pathogens but not against the necrotrophs[25], suggesting that the roles of DELLAs in plant immunity may vary depending upon the host species and the nature of the invading pathogen. In the light of this ambiguity, we were interested in exploring how DELLA proteins modulate plant antiviral immunity. It was reported that application of GA alleviated the symptoms of rice dwarf virus infection in rice plants, but there was no significant decline in relevant viral accumulation levels[27]. The significance of GA and DELLA hubs in plant immunity to pathogenic viruses is therefore still elusive.

In this study, we uncovered a common target, the rice DELLA protein SLR1, an important interactor with several different and unrelated viral effectors, including SRBSDV SP8, RSV P2 and RSMV M proteins. These viral proteins promoted specific degradation of SLR1 by physically coordinating the association of GA receptor OsGID1 with SLR1, thereby overcoming the SLR1-integrated broad-spectrum resistance to viral infection. In particular, these viral proteins efficiently contributed to the termination of JA signaling by disrupting SLR1-mediated JA signaling activation and recombination of the OsJAZ-OsMYC corepressor complex. These data provide a functional explanation for our previous results and suggest potential opportunities for strategies to protect rice crops from viral diseases.

## Results

### Several distinct viral proteins interact with SLR1

Following our previous demonstration that different rice viruses encode a class of functionally conserved transcriptional repressors, which broadly target several conserved host factors, including OsARF17, OsJAZs and OsMYC2/3[16,17], we wondered whether other crucial host factors or signaling pathways were commonly involved in viral infection. In preliminary experiments to identify target host factors of SRBSDV SP8 by screening a rice cDNA library, we obtained an interesting candidate SLR1, encoding a DELLA-like GRAS protein that acts as a master repressor of GA signaling[28,29]. We first carried out a subset of yeast two-hybrid (Y2H) assays to check the ability of SP8 to interact with SLR1, and found that only BD-SP8 and AD-SLR1 co-transformations were able to grow on SD-L-T-H-Ade selection plates, whereas yeast transformations carrying BD-SP8 and empty constructs failed to grow, implying that SLR1 physically interacted with SP8 in yeast cells (Fig. 1a). In further experiments SLR1 also interacted with other transcriptional repressors RBSDV P8, RSV P2 and RSMV M protein in yeast cells (Fig. 1a). Mapping of the functional domains using defined protein fragments of SLR1 revealed that the region containing the GRAS element, but not DELLA, mediated the interactions with these distinct viral proteins (Fig. 1a and Supplementary Fig. 1a–c).

Bimolecular fluorescence complementation (BiFC) assays were then used to test whether these reactions also occurred in vivo. When SP8-nYFP, P2-nYFP or M-nYFP were transiently expressed with SLR1-cYFP in Nicotiana benthamiana leaves by agro-infiltration, there was strong reconstitution of YFP fluorescence but not in the negative controls (Fig. 1b and Supplementary Fig. 1d), thus providing preliminarily evidence that SLR1 binds specifically and directly to these distinct viral proteins in planta. As confirmation of these in vivo interactions, we next conducted coimmunoprecipitation (Co-IP) assays, in which the SLR1-flag fusion protein was transiently co-expressed with SP8-myc, P2-myc or M-myc by agro-infiltration. All of the tested viral proteins were successfully precipitated by SLR1-flag, but there were no bands in GFP-flag negative control combinations (Fig. 1c–e). Together, these results demonstrate that SLR1 is a conserved interaction partner via its GRAS domain with the distinct viral proteins SP8, P2 and M.

### SP8 and RSV P2 provoke rapid degradation of SLR1

Because SP8 and P2 are encoded by very different viruses (respectively, SRBSDV, a double-stranded RNA virus, and RSV, a negative-stranded RNA virus) that occur frequently in rice fields, we chose them in the first instance to explore their functional relationship with SLR1. We first examined the subcellular localization of SP8-Ven (Venus, a yellow fluorescence protein) and SLR1-Tur (mTurquoise2, a cyan fluorescence protein) inside N. benthamiana cells. Surprisingly, the numbers of nuclear bodies formed by SLR1-Tur were dramatically decreased in the presence of SP8-Ven, compared with the single SLR1-Tur (Supplementary Fig. 2a). In line with this observation, western blot analysis showed that less SLR1 protein accumulated in SP8-Ven samples (Supplementary Fig. 2b), but this phenomenon was abolished in the presence of MG132, a 26S proteasome inhibitor (Supplementary Fig. 2a, b). However, at the transcriptional level, no major perturbation in SLR1 expression was observed in these different samples (Supplementary Fig. 2c), raising the possibility that viral proteins contribute to the destabilization of SLR1 by direct physical interaction.

To test this hypothesis, we transiently infiltrated SLR1-flag with equal proportions of SP8-myc, P2-myc or Gus-myc into N. benthamiana leaves. Immunoblotting analysis revealed that the accumulation of SLR1-flag in the presence of SP8-myc or P2-myc was only about 10% of that in control Gus-myc treated samples. MG132 treatment rescued the amount of SLR1-flag to approximately 70% (Fig. 2a, b), confirming that viral proteins SP8 and P2 indeed affected the stability of SLR1 though the 26 S proteasome pathway. In vitro time-course degradation experiments were then done to confirm the role of these viral proteins in destabilizing SLR1. We first expressed and purified His-SP8, His-P2 protein and His control from Escherichia coli in a cell-free system, endogenous SLR1 degraded more quickly when co-incubated with His-SP8 or His-P2 than in the presence of the His controls (Fig. 2c, d),

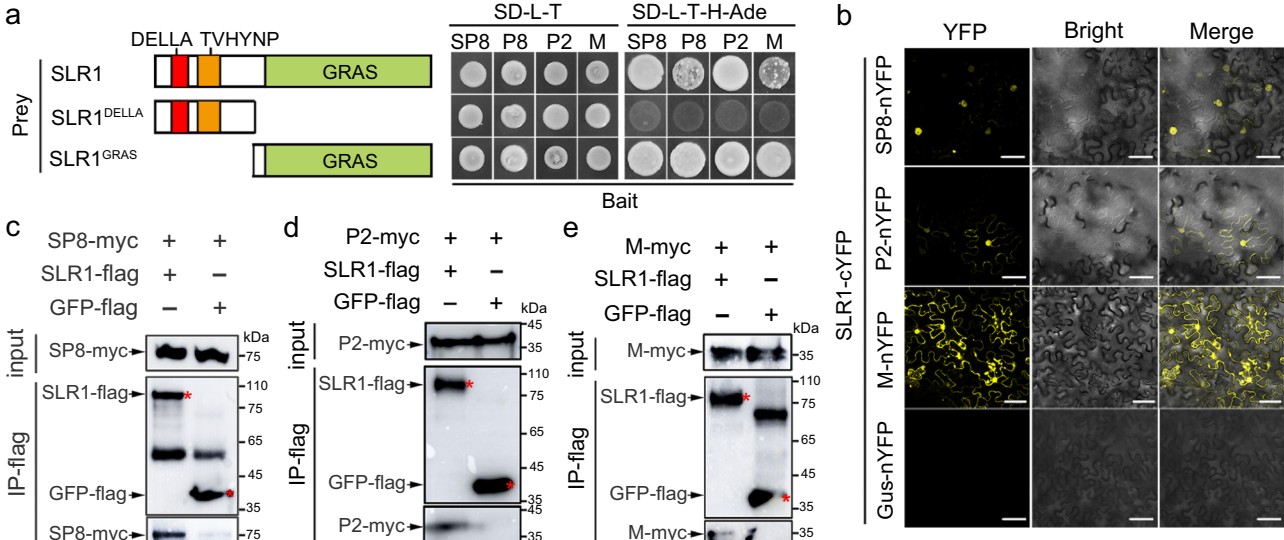

**Fig. 1 | GA-independent interactions between distinct viral proteins and SLR1.**
**a** Schematic diagrams of SLR1 and its deletion mutants and their interaction with distinct viral proteins (SRBSDV SP8, RBSDV P8, RSV P2 and RSMV M). The right panel shows that the conserved GRAS domain of SLR1 is required for interactions. In the Y2H system, viral proteins were fused with BD while SLR1 and its mutant derivatives were fused with AD yeast vectors. The different combinations of constructs transformed into yeast cells were grown on SD-L-T-H-Ade plates at 30 °C and photos were taken after 3 days. **b** BiFC assays confirming the interactions of SLR1 with viral proteins SP8 and RSV P2. SLR1-cYFP was agro-injected together with SP8-nYFP, P2-nYFP or Gus-nYFP into *Nicotiana benthamiana* leaves, and the samples were imaged by confocal microscopy at 48 hpi. Scale bar = 50 μm. Each experiment was repeated three times with similar results. Co-IP assay showing that SLR1 interacted with viral proteins SP8 (**c**), P2 (**d**) and M (**e**) in vivo. Total proteins were extracted from *N. benthamiana* leaves co-expressing SLR1-flag with SP8-myc, RSV P2-myc or RSMV M-flag, then precipitated with FLAG beads and probed with anti-flag and anti-myc antibodies for immunoblot analysis. The samples with GFP-flag served as negative control. Red asterisks indicate the specific band. Each experiment was repeated three times with similar results. Source data including uncropped scans of gels (**c–e**) are provided in the Source data file.

implying that SP8 and P2 proteins universally promote the degradation of SLR1 in *planta*. More importantly, analysis of the SLR1 levels in SP8-*ox* and *P2-ox* transgenic seedlings showed that the endogenous SLR1 protein level, but not the corresponding mRNA level, was significantly reduced in the plants expressing viral proteins compared to wild-type *Nipponbare* (*Nip*) plants (Supplementary Fig. 3).

SLR1 degradation or deficiency usually enhances plant sensitivity to GA treatment. Germinated seedlings of SP8-*ox* and *P2-ox* transgenic lines were grown in nutrient solutions with different concentrations of GA$_3$, and the second leaf sheaths were measured after 7 days, which is widely accepted as a good phenotypic measure of the GA response[30]. In the absence of GA$_3$, there were no significant differences in second leaf sheath length among *Nip*, SP8-*ox* and *P2-ox* lines. In the presence of low concentrations of GA$_3$ (0.1 μM and 1 μM), the second leaf sheath lengths of SP8-*ox* and *P2-ox* were nearly indistinguishable from wild-type *Nip* seedlings but at higher concentrations (2 μM, 5 μM and 10 μM) they were significantly longer than the controls (Fig. 2e–h). Together, these results indicated that SP8-*ox* and *P2-ox* transgenic plants were more sensitive to GA treatment because of the enhanced degradation of endogenous SLR1 caused by the viral proteins.

### SP8 and RSV P2 promote the interaction of OsGID1 with SLR1

Through its GA perception module, the GA receptor GID1 undergoes a conformational switch when GA is recognized, and this exposes the hydrophobic DELLA-binding surfaces, leading to the subsequent binding to DELLA[20,31]. Given the essential roles of OsGID1 in degradation of SLR1, we investigated whether SP8 and P2 proteins were themselves able to interact with OsGID1. In Co-IP assays with the OsGID1 receptor, both SP8-myc and P2-myc were successfully precipitated with OsGID1-flag, rather than GFP-flag (Fig. 3a, b). Y2H assays confirmed that OsGID1 interacted with both SP8 and RSV P2 in yeast cells (Supplementary Fig. 4a, b). Parallel in vitro pull-down results further showed that glutathione S-transferase (GST)–SP8 and GST-P2,

but not GST alone, pulled down MBP-His-OsGID1 and that these interactions were independent of GA$_3$ (Fig. 3c, d), demonstrating that viral proteins SP8 and P2 directly interacted with OsGID1 in vitro.

Since SP8 and P2 both physically associate with SLR1 and OsGID1, we wondered if these ternary interactions functionally affected the access of OsGID1 to SLR1 protein, thereby changing the stability of SLR1. We therefore carried out protein competition Co-IP assays using plant tissues co-expressing recombined SLR1 and OsGID1 in the presence or absence of viral proteins in *N. benthamiana*. The infiltrated leaves were treated with 50 μM MG132 at 24 hpi to avoid the rapid degradation of SLR1 triggered by the viral proteins, and were then harvested for immunoprecipitation by anti-flag beads 24 h later. Intriguingly, OsGID1-flag was precipitated by SLR1-GFP, and progressively with increasing quantities of SP8, without affecting the total endogenous protein levels (Fig. 3e), very similar results were obtained within RSV P2 protein (Fig. 3f). Following the competitive BiFC assays in leaves of *N. benthamiana*, the reconstituted YFP signals due to the interaction between SLR1-cYFP and OsGID1-nYFP were evident in the nucleus, and the fluorescence formed by the SLR1-OsGID1 complex was weaker but more aggregated in the presence of SP8 and P2 proteins (Supplementary Fig. 5), suggesting that SP8 and P2 likely function as linkers between SLR1 and OsGID1 receptor in *planta*.

To provide further biochemical evidence, we next designed competitive in vitro pull-down assays. Viral proteins SP8 and P2 interacted with both SLR1 and OsGID1 in the absence of GA$_3$ (Fig. 3c, d and Fig. 3g, h). In the presence of GA$_3$, the interaction between SLR1 and OsGID1 was enhanced by increasing amounts of either TF-His-SP8 or MBP-His-P2 (Fig. 3g, h), thus facilitating the functional complex formation of SLR1-SP8/P2-OsGID1. This further strengthens the hypothesis that SP8 and P2 proteins act as scaffolds to bring SLR1 physically adjacent to its receptor OsGID1, and that GA$_3$ recognition is indispensable to the functional degradation of SLR1. Collectively, these data have shown that the key viral proteins SP8 and P2 specifically promote

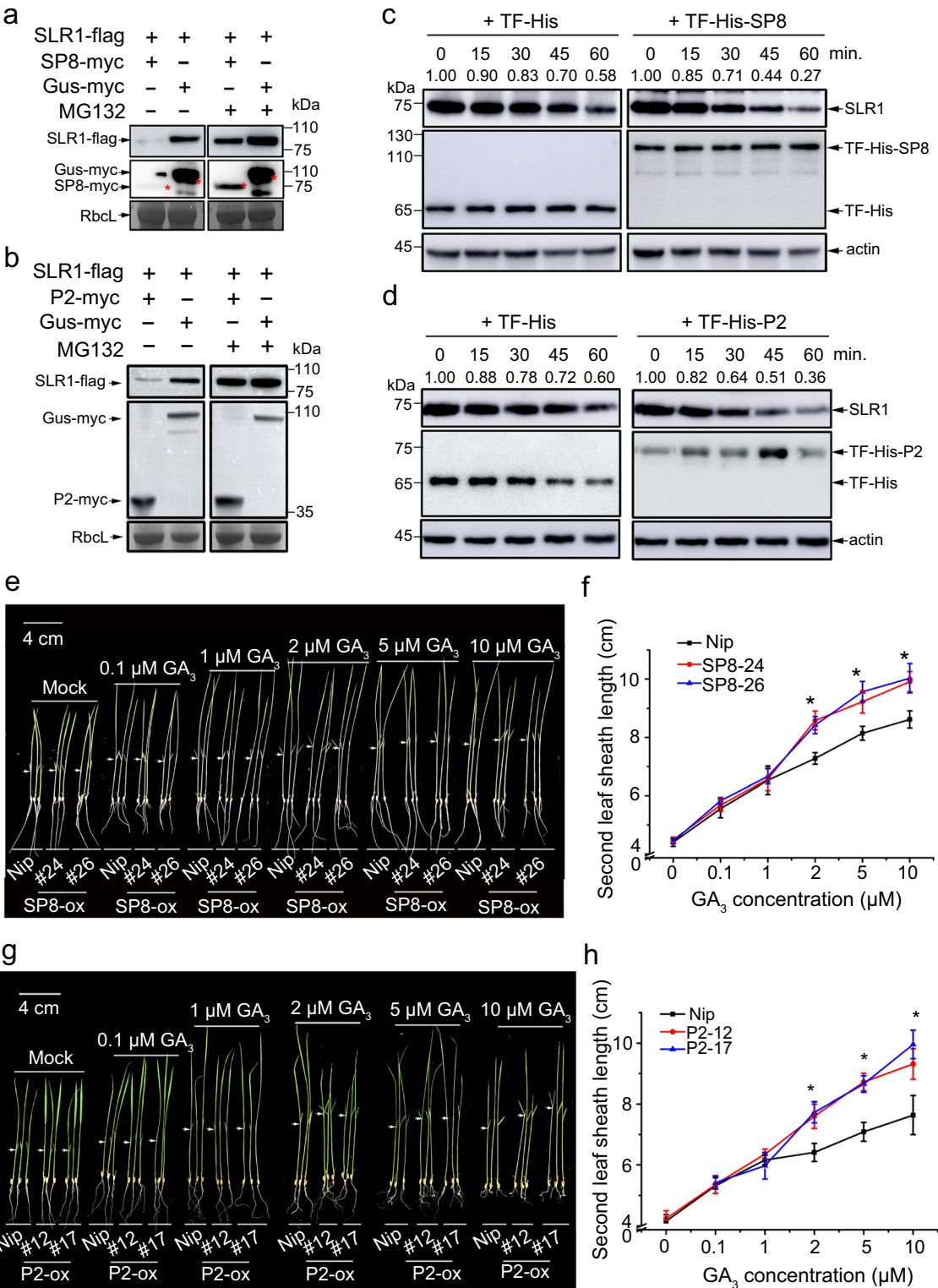

**Fig. 2 | SRBSDV SP8 and RSV P2 provoke rapid degradation of SLR1. a, b.** Effects of viral proteins RBSDV SP8 and RSV P2 on the accumulation of SLR1 in *N. ben-thamiana* leaves. The co-infiltrated leaves were treated with MG132 (50 μM) or DMSO at 24 hpi and then were harvested for western blotting 24 h later. RbcL was used as a loading control. Each experiment was repeated three times with similar results. **c, d.** In vitro degradation assay. Total protein extracted from *Nip* seedlings was incubated in 10 μM GA₃ with approximately 50 μg purified TF-His, TF-His-SP8 **(c)** or TF-His-P2 **(d)** protein from *E. coli* at 30 °C incubator for indicated times, and these samples were collected at the indicated times for western blot using anti-SLR1 and anti-His. Anti-actin was used as a loading control. Each experiment was

repeated three times with similar results. Phenotypes of *Nip*, SP8-*ox* **(e)** or *P2-ox* **(g)** seedlings treated with GA₃. Similar germinated seeds were planted in different concentrations of GA₃ (0, 0.1, 1, 2, 5, 10 μM) containing culture solution for about 7 d, *n* = 3 biologically independent replicates per genotype. All images were pho-tographed using a digital camera. Scale bar = 4 cm. Second leaf sheath lengths of *Nip*, SP8-*ox* **(f)** or *RSV P2-ox* **(h)** seedlings treated with GA₃. Values were obtained from *n* = 15 biologically independent plants. * at the top of columns indicate sig-nificant differences ($p < 0.05$) based on Fisher's least significant difference tests. Source data including uncropped scans of gels (**a**–**d**) and *p* values of statistic tests (**f** and **h**) are provided in the Source data file.

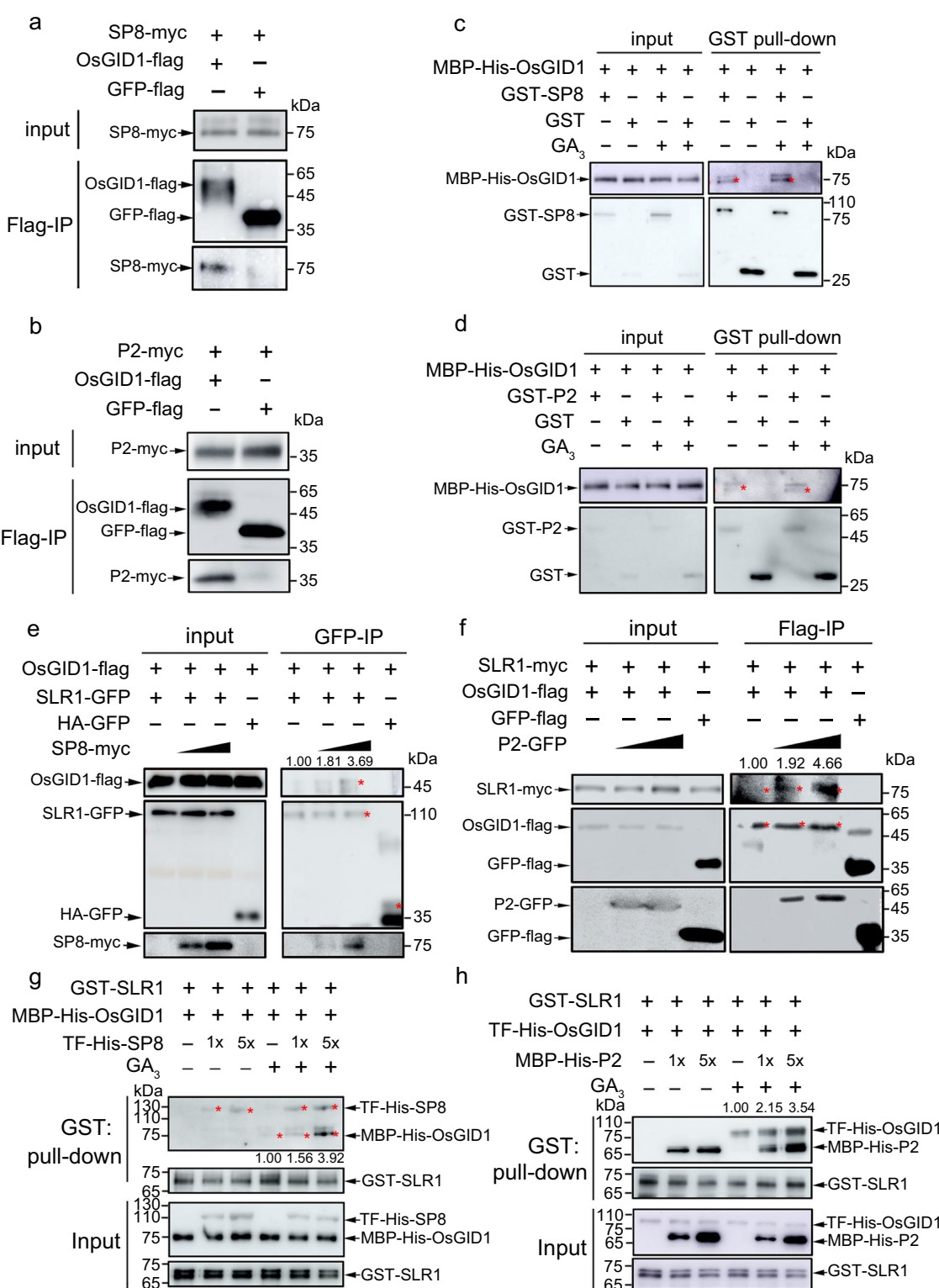

the association between SLR1 and OsGID1, resulting in the rapid degradation of SLR1.

## SLR1-mediates antiviral defense is repressed by viral proteins

Since the viral proteins accelerate SLR1 degradation, we next explored the potential effect of the GA signaling pathway on viral infection. The transgenic rice plants *SLR1-GFP*, overexpressing SLR1 fused with green fluorescent protein, and GA signaling mutant *RNAi-SLR1* in which the expression of SLR1 was knocked down by RNA interference[32], were inoculated with viruses[33]. After challenging with RSV, a representative negative single-stranded (ss) (-) RNA virus, *RNAi-SLR1* mutant plants exhibited more severe symptoms than the WT (*Lansheng, LS*), with discontinuous yellow stripes and necrotic streaks on the leaves. Conversely, much milder virus symptoms were observed in seedlings overexpressing *SLR1-GFP* (Fig. 4a). The severity of viral symptoms on rice leaves was scored as healthy without symptoms (N), typical yellow stripes (I) and curling or death (II) of young leaves[34]. Fewer *RNAi-SLR1* plants had no symptoms (Grade N) and more had Grade I and II of

**Fig. 3 | SP8 and RSV P2 promote interaction of OsGID1 with SLR1.** Co-IP analyses of the interactions between viral proteins SP8 **(a)** and RSV P2 **(b)** with OsGID1 in *N. benthamiana* leaves. Each experiment was repeated three times with similar results. **c, d.** Pull-down assays for analysis of the interaction between OsGID1 and SP8 **(c)** or P2 **(d)**. An equal amount of MBP-His-OsGID1 was incubated with immobilized GST and GST-SP8 **(c)** or GST-P2 **(d)** separately, and then the bound proteins were detected by Western blotting using anti-His and anti-GST antibodies. Each experiment was repeated three times with similar results. Co-IP assays showing that SP8 and RSV P2 promote interaction of OsGID1 with SLR1 *in planta*. OsGID1-flag and SLR1-GFP/SLR1-myc were transiently infiltrated using Agrobacterium together with/without increasing SP8-myc **(e)** or P2-GFP **(f)** in leaves of *N. benthamiana*, while leaves expressing HA-GFP or GFP-flag were used as negative controls.

Cultures were pelleted to a final $OD_{600}$ of 0.5, increasing amounts of SP8 and P2 following agrobacterium infection with final $OD_{600}$ = 0.5 or $OD_{600}$ = 1.0, respectively. Proteins were immunoprecipitated with Flag- paramagnetic beads. The red asterisks point to the specific band. The co-infiltrated leaves were treated with MG132 (50 μM) or DMSO at 24 hpi and then harvested for coimmunoprecipitation 24 h later. Each experiment was repeated three times with similar results. In vitro pull-down assays showing the effects of SP8 **(g)** and RSV P2 **(h)** on the activation of the interaction between OsGID1 and SLR1. The indicated proteins were purified from *E. coli* and pulled down by GST beads. Immunoblots were performed using anti-GST and anti-His antibodies to detect the associated proteins. Each experiment was repeated three times with similar results. Source data including uncropped scans of gels (a–h) are provided in the Source data file.

young leaves, whereas fewer *SLR1-GFP* plants had severe disease symptoms (grade II) than in the WT (Fig. 4b). Consistently, the levels of RSV coat protein (CP) were significantly lower in *SLR1-GFP* lines but accumulated significantly more in *RNAi-SLR1* mutants (Fig. 4c, d). Together, these results suggested that SLR1 plays vital roles in rice antiviral defense against RSV infection.

To test whether SLR1-mediated resistance could be compromised by these distinct viral proteins, we compared the effect of SLR1 on RSV infection in the presence of SP8 or P2. In repeated trials, more seedlings of SP8-ox and P2-ox lines displayed disease symptoms and there was more severe curl or death when infected by RSV (Supplementary Fig. 6a, b). Viral RNA and CP were increased in SP8-ox and P2-ox lines compared with the *Nip* controls (Supplementary Fig. 6c, d), indicating the functional significance of SP8 and P2 in impeding defense against RSV. To further confirm this, we crossed SP8-ox, P2-ox and their background *Nip* with *SLR1-GFP*, and ultimately obtained two homozygous lines for each transgenic combination. *SLR1-GFP*/SP8-ox and *SLR1-GFP*/P2-ox plants were taller, had more panicles and seed germinated more quickly than *SLR1-GFP*/*Nip* controls (Supplementary Fig. 7) presumably because of the decreased SLR1 protein content. Consistent with these phenotypes, *SLR1-GFP*/SP8-ox and *SLR1-GFP*/P2-ox lines had more severe stunting when infected by RSV (Fig. 4e), and higher percentages of typical disease symptoms (Grade I and II on their leaves) than *SLR1-GFP*/*Nip* plants (Fig. 4f). Consistent with this, RSV RNAs and CP accumulated to significantly higher levels in *SLR1-GFP*/SP8-ox and *SLR1-GFP*/P2-ox lines than in *SLR1-GFP*/*Nip* plants (Fig. 4g, h). Together, these results indicate that SLR1 contributes to rice resistance to RSV, while SP8 and P2 directly compromise SLR1-mediated antiviral immunity, conferring a selective advantage to RSV multiplication.

To further substantiate the significance of SLR1 against viruses in general, similar experiments were done using SRBSDV, a representative double-stranded RNA (dsRNA) virus. Similar to the results with RSV, transgenic *SLR1-GFP* plants had milder dwarfing symptoms of SRBSDV, with reduced disease incidence, decreased amounts of SRBSDV RNAs (*S2, S4* and *S6*) and corresponding CP (SRBSDV P10) protein accumulation (Fig. 5a–d), while SLR1-knockdown rice plants *RNAi-SLR1* and *SLR1-GFP*/SP8-ox, *SLR1-GFP*/P2-ox cross lines were consistently more sensitive to SRBSDV infection with more severe symptoms, higher levels of viral RNAs (*S2, S4 and S6*) levels and greater CP accumulation (Fig. 5e–h). Collectively, the evidence strongly implies that SLR1-mediated broad-spectrum antiviral resistance is subverted by viral proteins SP8 and P2, thus allowing the establishment of infection by different RNA viruses.

**Viral proteins restrict SLR1 to activate JA antiviral signaling**
Our recent work showed that the viral proteins SP8 and P2 negatively modulate JA signaling by cooperating with OsJAZ repressors to repress the transcriptional activation of OsMYC2/3[17], and other reports show that DELLA proteins participate in the JA pathway by directly interacting JAZ proteins[35,36]. We therefore next examined how SLR1 crosstalks with the JA signaling pathway during viral infection. To identify the intriguing ternary relationship involving SLR1, the JA signaling

components (OsJAZ, OsMYC2/3) and viral proteins SP8 and P2, we first screened for interactions between SLR1 and JA components in Y2H assays. We found that SLR1 consistently associated with several OsJAZ family proteins (OsJAZ3, OsJAZ4, OsJAZ9 and OsJAZ12) in yeast cells (Supplementary Fig. 8a), and these interactions were verified by BiFC and Co-IP assays in *N. benthamiana* based transient systems (Supplementary Fig. 8b–f). Consistent with these findings, Co-IP assays showed that SLR1 had a strong affinity for OsMYC2 or OsMYC3, while no bands appeared in GFP-flag negative combinations (Supplementary Fig. 8g, h). Together, these results strongly indicate that SLR1 widely associates with OsJAZ members and OsMYC2/3.

Because it is currently believed that DELLA proteins may compete with MYC2 for binding to JAZ1 in *Arabidopsis*[36], we next investigated whether this also occurs in monocotyledonous rice plants. Confocal fluorescence imaging showed that increasing the quantity of SLR1 progressively decreased the YFP fluorescence signals of the OsJAZ9-cYFP/OsMYC3-nYFP combinations (Supplementary Fig. 9), suggesting that SLR1 efficiently triggers the dissociation between OsJAZ9 and OsMYC3. Consistent with this observation, the growth inhibition effect of methyljasmonate (MeJA) was dramatically increased in the *SLR1-GFP* seedlings, but significantly decreased in the *RNAi-SLR1* mutants in comparison with the *LS* control (Supplementary Fig. 10), further supporting the view that SLR1 participates in JA-mediated growth inhibition. These results collectively demonstrate that SLR1 positively modulates JA signaling by competing with the binding of OsJAZ proteins to OsMYC2/3 in rice.

We next dissected the impact of viral proteins on the specific association between SLR1 and OsJAZ9 or OsMYC3. As the viral proteins and OsJAZ proteins all interacted with the GRAS domain of SLR1 (Supplementary Fig. 11), we designed competitive Co-IP experiments. In cells co-expressing viral proteins with SLR1 and OsJAZ9, the infiltrated areas were pre-treated with 50 μM MG132 to stabilize SLR1 and then sampled for immunoprecipitation with anti-flag beads. Notably, the interaction between SLR1 and OsJAZ9 was dissociated by increasing amounts of SP8 or P2 protein (Fig. 6a, b). Consistent with this, there was a similar effect preventing inhibition and hijacking OsMYC3 (Fig. 6c, d). These results demonstrate convincingly that viral proteins SP8 and P2 coordinately block the ability of SLR1 to bind with JA signaling components OsJAZ or OsMYC3. We therefore next investigated the effect of viral proteins on the SLR1-OsJAZ-OsMYC3 complex. As shown in Fig. 6e, f, under normal conditions SLR1 impaired the ability of OsJAZ9 to bind with OsMYC3 but in the presence of SP8 or P2, the ability of OsJAZ9 to bind with OsMYC3 was enhanced. These results show that expression of SP8 or P2 protein interferes with the inhibitory effect of SLR1 on the OsJAZ9-OsMYC3 interaction, resulting in the recombination of OsJAZ9-OsMYC3. Together, these data show that viral proteins SP8 or P2 disrupt the involvement of SLR1 with JA signaling.

To further confirm the physiological relevance of the targeted degradation of SLR1 by viral proteins and the subsequent compromise of JA signaling, we treated the transgenic rice plants *SLR1-GFP/Nip*, *SLR1-GFP*/SP8-ox and *SLR1-GFP*/P2-ox with 0.1 μM and 1 μM

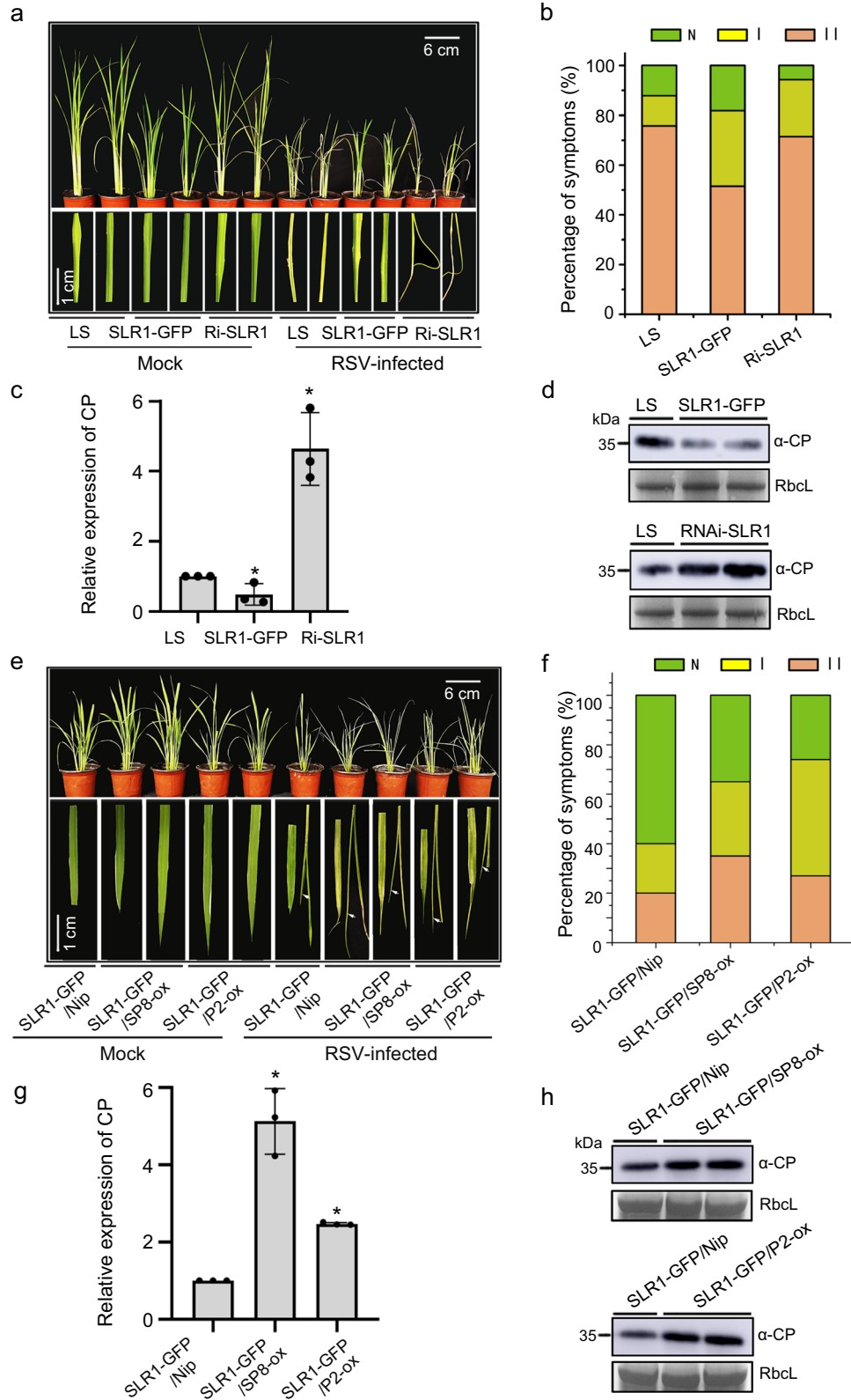

MeJA for 7 days under hydroponic conditions. Expression of SP8 or P2 effectively subverted the phenotype of *SLR1-GFP* lines making them less sensitive to JA-mediated inhibition of root growth (Fig. 6g, h). Collectively, these results validate the biological significance of the functional links between SLR1 and JA response and confirm that the viral proteins restrict the ability of SLR1 to integrate with JA signaling.

## RSMV M protein also manipulates SLR1

Since the M protein of RSMV (a single-stranded RNA virus belonging to the genus *Cytorhabdovirus*) also interacted with SLR1 (Fig. 1a, b, e), we investigated whether M protein acts as a conserved pathogenic effector to manipulate SLR1 degradation in a similar way to SRBSDV SP8 and RSV P2. The results showed that M protein does indeed interact with OsGID1 (Fig. 7a). As expected, SLR1 was obviously

**Fig. 4 | SLR1 confers resistance to RSV infection in rice. a** Plants and leaves of *LS*, *SLR1-GFP* and *RNAi-SLR1* 20 d after inoculation with RSV. The areas of typical yellow stripes and curl or death of the young leaves represent the degree of disease symptoms. Values were obtained from *n* = 30 biologically independent plants, *n* = 3 biologically independent replicates per genotype. Scale bars = 6 cm (upper panel) and 1 cm (lower panel). **b** Disease incidence and grades of symptoms in *LS*, *SLR1-GFP* and *RNAi-SLR1* 20 d after inoculation with RSV. **c** Results of qRT-PCR showing the relative mRNA levels of RSV *CP* in RSV-infected *LS*, *SLR1-GFP* and *RNAi-SLR1* rice plants. * at the top of columns indicate significant differences (p < 0.05) based on Fisher's least significant difference tests. **d** The accumulation of RSV CP protein in RSV-infected *LS*, *SLR1-GFP* and *RNAi-SLR1* rice plants by western blot. RbcL serves as the loading control. **e** Plants and leaves of *SLR1-GFP/Nip*, *SLR1-GFP/*SP8*-ox* and *SLR1-GFP/P2-ox* 20 d after inoculation with RSV. The areas of typical yellow stripes and curl or death (highlighted as white arrowheads) of the young leaves represent the degree of disease symptoms. Values were obtained from n = 30 biologically independent plants, *n* = 3 biologically independent replicates per genotype. Scale bars = 6 cm (upper panel) and 1 cm (lower panel). **f** Disease incidence and grades of symptoms in transgenic rice plants *SLR1-GFP/Nip*, *SLR1-GFP/*SP8*-ox* and *SLR1-GFP/P2-ox* 20 d after inoculation with RSV. **g** Results of qRT-PCR showing the relative expression levels of RSV *CP* in RSV-infected *SLR1-GFP/Nip*, *SLR1-GFP/*SP8*-ox* and *SLR1-GFP/P2-ox* rice plants. * at the top of columns indicate significant differences (*p* < 0.05) based on Fisher's least significant difference tests. **h** Western blot showing the accumulation of RSV CP protein in RSV-infected *SLR1-GFP/Nip*, *SLR1-GFP/*SP8*-ox* and *SLR1-GFP/P2-ox* rice plants. RbcL serves as the loading control. Source data including uncropped scans of gels (**d** and **h**) and *p* values of statistic tests (**c** and **g**) are provided in the Source data file.

degraded when co-expressed with M protein (but not with GUS negative control) in *N. benthamiana*, while treatment with MG132 inhibitor largely rescued the accumulation (Fig. 7b). Kinetic analysis of SLR1 degradation in a cell-free system also gave very similar results with accelerated decay of SLR1 in the presence of His-M compared with the His control (Supplementary Fig. 12). Similarly, endogenous SLR1 was more rapidly degraded in extracts from *M-ox* rice plants than in those from *Nip* plants (Fig. 7c), confirming that RSMV M was also responsible for SLR1 destabilization. Importantly, considering the SLR1-stabilizing effect of M protein, we further analyzed the GA sensitivity of *M-ox* transgenic plants. The second leaf sheath of *M-ox* lines was distinctly longer in response to GA$_3$ treatment, corresponding with compromised accumulation of endogenous SLR1 in *M-ox* plants (Fig. 7d, e). These results collectively suggest that RSMV M protein favors GA sensitivity by promoting the degradation of SLR1.

To further explore the function of SLR1 in RSMV infection, we compared the symptoms of RSMV infection on the transgenic rice plants *SLR1-GFP* and *RNAi-SLR1* with *LS*. The *SLR1-GFP* lines exhibited slight dwarfing, striped mosaicism and were stiff, but the mosaic symptoms occurred earlier in *RNAi-SLR1* mutants, and symptoms were more severe with crinkling, twisting and the appearance of wax or swellings on the young leaves later (Fig. 7f). The *RNAi-SLR1* mutants had up to 74% plants infected with significantly fewer in *SLR1-GFP* lines (35%) and *LS* wild-type plants (56%) (Fig. 7g). The levels of RSMV RNAs (*M* and *N*) were also always lower in the *SLR1-GFP* lines and higher in *RNAi-SLR1* mutants than *LS* plants (Fig. 7h), and similar results were obtained in western blots to detect the protein levels using an anti-M antibody (Fig. 7i). It appears therefore that SLR1 contributes to rice antiviral defense against the new cytorhabdovirus RSMV, in a manner similar to its role against SRBSDV SP8 and RSV P2.

In summary, these findings favor a potential working model that several different viral proteins, including SRBSDV SP8, RSV P2 and RSMV M, commonly promote the degradation of SLR1 by boosting its affinity with GA receptor OsGID1. Meanwhile, these viral proteins diminish the ability of SLR1 to activate JA signaling by disassociating SLR1 from the OsJAZ-OsMYC2/3 complex, and thereby repress SLR1-mediated board-spectrum antiviral defense (Fig. 8).

## Discussion

Plants have various sophisticated strategies that equip them to overcome the threats posed by pathogens, including the deployment of phytohormones as part of the host innate immunity system against viruses[37–39]. Our earlier reports have shown that viral infection can disrupt hormonal networks, and in particular that jasmonate (JA) and auxin signaling were synergistically involved in resistance to RBSDV[40,41], while enhanced brassinosteroid (BR) and abscisic acid (ABA) pathways made plants more susceptible to RBSDV infection[42,43]. However, little is yet known about any antiviral function of gibberellin (GA) and its DELLA hub. Prior to this study, investigation of the GA pathway in plant-virus interactions has mainly focused on its role in viral-induced disease symptom development[27,44]. DELLA proteins are

the master negative regulators of GA signaling and regulate plant resistance against fungal and bacterial pathogens in different ways. For example, DELLA member RGL3 positively regulates plant resistance to the necrotrophic fungus *Botrytis cinerea* but not to the hemibiotrophic bacterial pathogen *Pseudomonas syringae* in *Arabidopsis thaliana*[26]. In contrast, DELLA protein SLR1 enhances rice defense against the hemibiotrophic pathogens *Magnaporthe oryzae* and *Xanthomonas oryzae pv. oryzae* but not against the necrotrophs *Rhizoctonia solani* and *Cochliobolus miyabeanus*[25]. Although the DELLA proteins have been reported to be central players in defense regulatory networks during tissue invasion, their role in the modulation of virus infections has been little studied. A recent study has shown that tobamovirus-cg coat protein (CgCP) manipulates DELLA proteins and in turn negatively modulates the salicylic acid-mediated defense pathway during *A. thaliana* infection[45]. Very recently it has been reported that geminivirus C4 proteins also influence viral pathogenicity by stabilizing the DELLA protein to promote viral infection and symptom development in *N. benthamiana*[46]. However, there is even less information about the significance of DELLA proteins in resistance to plant viruses in monocotyledonous crop plants and particularly in the model crop rice (*Oryza sativa*). In this study, the relationship between rice DELLA protein SLR1 and plant viruses was investigated in detail. We have observed that overexpression of SLR1 makes rice plants resistant to several different viruses, including the dsRNA virus SRBSDV and the ssRNA viruses RSV and RSMV, whereas the *RNAi-SLR1* mutant was susceptible to rice viral infection (Figs. 4a, 5a and 7e). These results suggest that SLR1 modulates board-spectrum antiviral defense in rice.

Host-virus interactions involve diverse and dynamic regulatory networks, which allow viruses to alter plant defensive signaling to favor infection[47,48]. Previous findings in our laboratory support the view that several different plant RNA viruses (dsRNA viruses SRBSDV/RBSDV belonging to the genus *Fijivirus* and ssRNA viruses RSV belonging to the genus *Tenuivirus* and RSMV belonging to the genus *Cytorhabdovirus*) coordinately manipulate the JA signaling pathway to facilitate viral infection and vector feeding. These distinct viruses have independently evolved viral transcriptional repressors, which directly interact with and disrupt the transcriptional activation complex OsMYC2/3-OsMED25, and repress JA signaling[17]. In addition to targeting JA signaling, these distinct viral proteins interfere with the same key component of auxin signaling, OsARF17, to inhibit its mediated antiviral response[16]. Thus, manipulation of critical defensive hormonal pathways appears to be a vital and conserved counter-defense strategy to help viruses overcome obstacles to systemic infection. Here, we document a common pathogenicity strategy adopted by different viruses in rice. In particular, we elucidate a detailed mechanism by which four different viral effectors, SP8/RBP8, RSV P2 and RSMV M proteins, counteract host defenses by interacting with both SLR1 and its receptor OsGID1. This viral hijacking of host targets promotes the degradation of SLR1 and attenuates the broad-spectrum SLR1-mediated antiviral defense in rice plants.

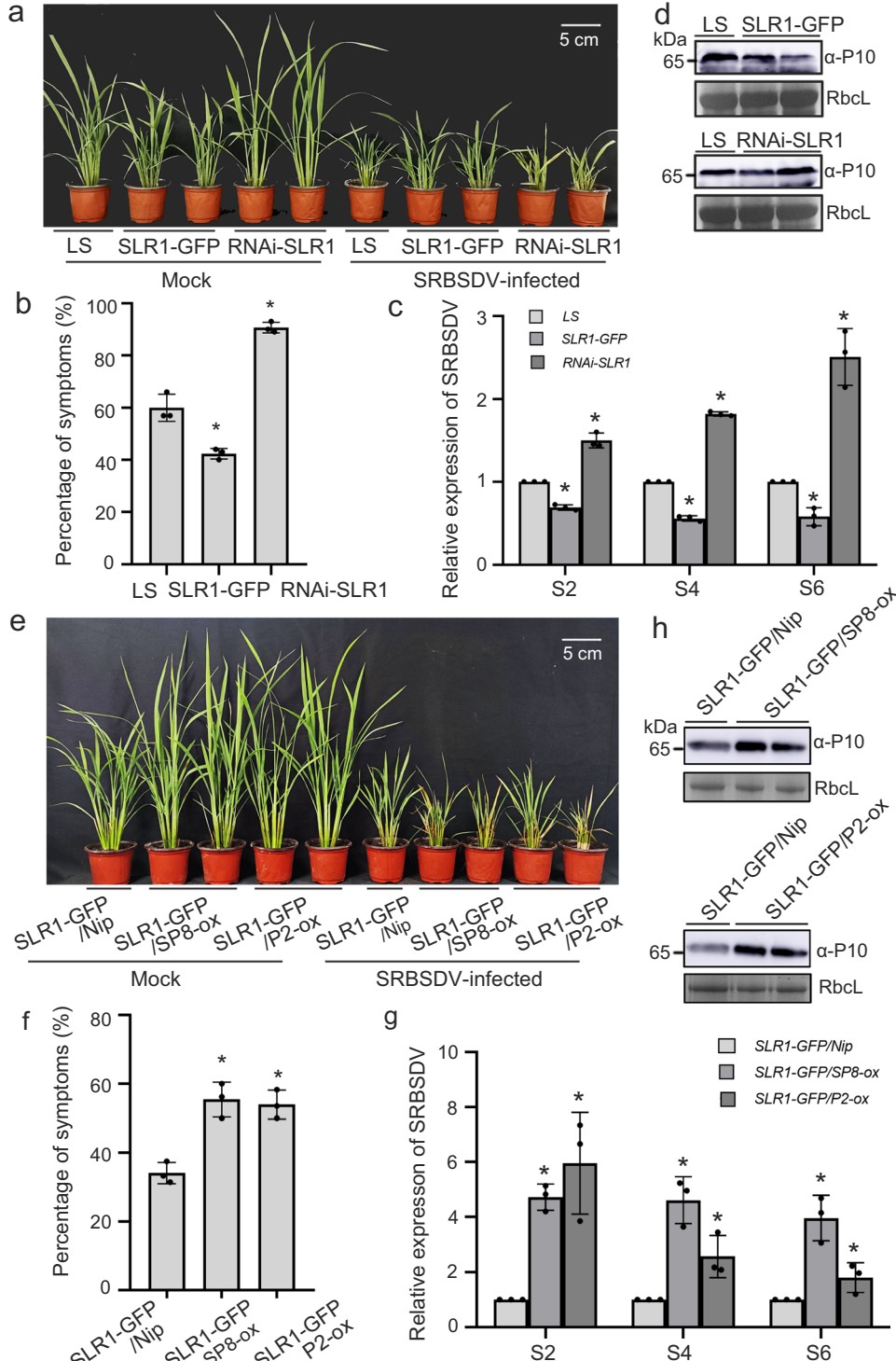

**Fig. 5 | SLR1 confers resistance to SRBSDV infection in rice. a** Symptoms on *LS*, *SLR1-GFP* and *RNAi-SLR1* plants 20 d after inoculation with SRBSDV. Values were obtained from *n* = 30 biologically independent plants, *n* = 3 biologically independent replicates per genotype. Scale bars=5 cm. **b** The percentages of SRBSDV symptomatic rice plants (%) in *LS, SLR1-GFP* and *RNAi-SLR1*. * at the top of columns indicate significant differences (*p* < 0.05) based on Fisher's least significant difference tests. **c** Results of qRT-PCR showing the relative expression of SRBSDV RNAs (*S2, S4* and *S6*) in infected *LS, SLR1-GFP* and *RNAi-SLR1* rice plants. * at the top of columns indicate significant differences (*p* < 0.05) based on Fisher's least significant difference tests. **d** The accumulation of SRBSDV CP protein in infected *LS, SLR1-GFP* and *RNAi-SLR1* rice plants by western blot. RbcL serves as the loading control. **e** Symptoms on transgenic *SLR1-GFP/Nip*, SLR1-GFP/SP8-ox and *SLR1-GFP/ P2-ox* plants 20 d after inoculation with SRBSDV. Values were obtained from *n* = 30

biologically independent plants, *n* = 3 biologically independent replicates per genotype. Scale bars=5 cm. **f** The percentages of SRBSDV symptomatic rice plants (%) in transgenic *SLR1-GFP/Nip*, SLR1-GFP/SP8-ox and *SLR1-GFP/P2-ox*. * at the top of columns indicate significant differences (p < 0.05) based on Fisher's least significant difference tests. **g** Results of qRT-PCR showing the relative expression levels of SRBSDV RNAs (*S2, S4* and *S6*) in *SLR1-GFP/Nip*, SLR1-GFP/SP8-ox and *SLR1-GFP/P2-ox* rice plants. * at the top of columns indicate significant differences (*p* < 0.05) based on Fisher's least significant difference tests. **h** Western blot showing the accumulation of SRBSDV CP protein in infected *SLR1-GFP/Nip*, SLR1-GFP/SP8-ox and *SLR1-GFP/P2-ox* rice plants. RbcL serves as the loading control. Source data including uncropped scans of gels (**d** and **h**) and *p* values of statistic tests (**c** and **g**) are provided in the Source data file.

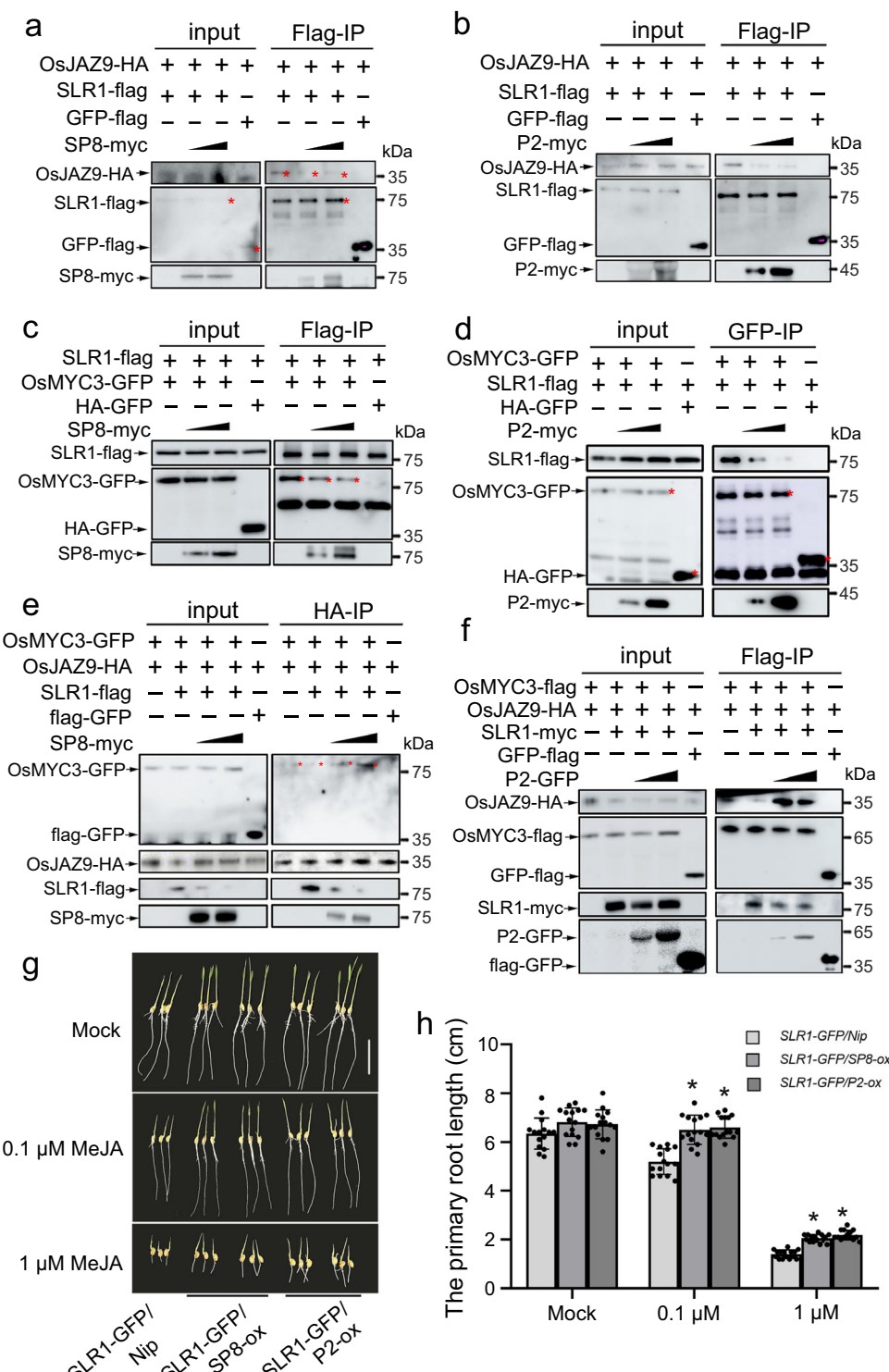

SLR1 functions as a hub that integrates a complicated regulatory network[18,49] and the molecular interactions between SLR1 and other host components are therefore critical to the outcome of any infection process. The existing literature indicates that DELLA proteins modulate JA signaling in the dicot plant *Arabidopsis*[35,36]. We now show in the monocot rice plant that SLR1 specifically interacts with multiple OsJAZ proteins and OsMYC2/3 transcription factors, and strongly impedes the combination of OsJAZ-OsMYC to activate JA signaling. As one of the most critical defense phytohormones, JA is involved in host defense by fine-tuning the expression of pathogenesis-related (*PR*) genes, thus inducing the initial immunity response to viral infection[39]. It was

recently shown that there is crosstalk between JA-mediated signaling and RNA silencing, and that accumulating JA promotes rice antiviral defense by inducing *AGO18* expression[34]. Our study provides strong evidence of the key role of SLR1 in JA-mediated broad-spectrum defense against distinct virus infections. The role of SLR1 and JA signaling in reducing symptoms and viral multiplication may be mediated by the activation of downstream *PR* gene expression and basal immunity or by integrating a multilayer defense with RNAi-mediated antiviral responses. Further identification of the biochemical mechanisms that drive broad-spectrum antiviral resistance in rice crops by manipulating SLR1 will enrich our knowledge about host-virus

**Fig. 6 | Viral proteins restrict SLR1 to activate JA antiviral signaling. a, b** Protein competition analyzed by Co-IP assays in vivo. OsJAZ9-HA and SLR1-flag were infiltrated using *Agrobacterium* together with increasing amounts of SP8-myc **(a)** or P2-myc **(b)** in leaves of *N. benthamiana*, GFP-flag serves as negative control. Cultures were pelleted to a final $OD_{600}$ of 0.5, increasing amounts of SP8 and P2 following agrobacterium infection with final $OD_{600} = 0.5$ or $OD_{600} = 1.0$, respectively. The co-infiltrated leaves were treated with MG132 (50 μM) or DMSO at 24 hpi and then harvested 24 h later for coimmunoprecipitation with Flag-tag paramagnetic beads. Each experiment was repeated three times with similar results. **c, d** Protein competition analyzed by Co-IP assays in vivo. OsMYC3 and SLR1 were infiltrated using *Agrobacterium* together with increasing amount of SP8-myc **(c)** or P2-myc **(d)** in leaves of *N. benthamiana*, HA-GFP used as negative controls. Cultures were pelleted to a final $OD_{600}$ of 0.5, increasing amounts of SP8 and P2 following agrobacterium infection with final $OD_{600} = 0.5$ or $OD_{600} = 1.0$, respectively. The co-infiltrated leaves were treated with MG132 (50 μM) or DMSO at 24 hpi and then harvested 24 h later for coimmunoprecipitation with Flag-tag paramagnetic beads. Each experiment was repeated three times with similar results. **e, f.** Protein competition

analyzed by Co-IP assays in vivo. OsMYC3-GFP and OsJAZ9-HA were infiltrated using *Agrobacterium* together with SLR1-flag in leaves of *N. benthamiana*. Increasing amounts of SP8-myc **(e)** or **(f)** were mixed to degrade SLR1 and GFP-flag was used as negative control. Cultures were pelleted to a final $OD_{600}$ of 0.5, increasing amounts of SP8 and P2 following agrobacterium infection with final $OD_{600} = 0.5$ or $OD_{600} = 1.0$, respectively. Proteins were harvested at 48 hpi and then immunoprecipitated with corresponding HA-tag **(e)** or Flag-tag **(f)** paramagnetic beads. Each experiment was repeated three times with similar results. **g** Phenotypes of *SLR1-GFP/Nip*, SLR1-GFP/SP8-ox and *SLR1-GFP/P2-ox* seedlings grown for 7 days on rice nutrient solutions with different concentrations of MeJA (0, 0.1, 1 μM), $n = 3$ biologically independent replicates per genotype. All images were photographed using a digital camera. Scale bar = 3 cm. **h** Root lengths of *SLR1-GFP/Nip*, SLR1-GFP/SP8-ox and *SLR1-GFP/P2-ox* seedlings. Error bars represent SD, * at the top of columns indicate significant differences (p < 0.05) based on Fisher's least significant difference tests. Values were obtained from n = 15 biologically independent plants. Source data including uncropped scans of gels (**a–f**) and *p* values of statistic tests (**h**) are provided in the Source data file.

relationships and help design high efficiently strategies to protect viral damage.

Since our previous study had shown that viral proteins interacted with the key components of JA signaling (OsJAZs and OsMYC2/3), we investigated the interplay between SLR1 and the JA signaling pathway during viral infection. As shown in Fig. 6, the diverse viral proteins SP8, P2 and M all disrupt the association of SLR1 with OsJAZ and OsMYC, and promote the recombination of OsJAZ and the OsMYC complex, resulting in the inactivation of JA signaling. This is a significant advance in our understanding, showing how these virulence effectors coordinately diminish SLR1 and how this function in concert with the JA signaling pathway to the advantage of the virus. This study together with our previous research thus shows that several different viral effectors convergently target dozens of common host defensive-related proteins, including the essential components in JA, Auxin and GA signaling, indicating that hijacking host components in phytohormone pathways is a common counter-defense strategy in viral pathogenesis in the monocotyledonous crop rice.

## Methods

### Plant materials and growth conditions
Rice (*Oryza sativa* ssp *japonica*) cv *Nipponbare* (*Nip*) and cv *Lansheng* (*LS*) were used in this study. Related GA transgenic rice plants (including the mutant *RNAi-SLR1* and the overexpression line *SLR1-GFP*[32], and viral-related transgenic rice plants (*SP8-ox* and *P2-ox* derived from the wild-type *Nip*)[16,17] were collectively used in our research. In this study, *SLR1-GFP* was crossed with *Nip*, *SP8-ox* and *P2-ox*, and two independent double overexpression lines were obtained. All these rice plants were grown in a greenhouse at 28–30 °C or in the field. *N. benthamiana* plants were grown in a growth chamber at 25 °C and with a 16-h-light/8-h-dark cycle.

### Yeast two-hybrid analysis
For Y2H screening, the full-length of SP8 was cloned into the pGBKT7 vector, and used as a bait to screen the rice cDNA library according to the manufacturer's protocol (Clontech). The isolated colonies were first selected from SD/-Leu/-Trp/-His, and then transferred onto medium plates SD/-Ade/-Leu/-Trp/-His/-x-α-gal. To confirm the interaction between SP8 and SLR1, the pGBKT7-SP8 and pGADT7-SLR1 constructs were co-transformed into yeast strain AH109 and the transformants were successively cultivated on SD/-Leu/-Trp and SD/-Leu/-Trp/-His/-Ade plates. All experiments were repeated three times with similar results.

### Bimolecular fluorescence complementation (BiFC)
The transient expression binary vectors constructed for BiFC assays were all derived from the *pCV1300* plasmid. The indicated DNA

fragments, including the full-length of SLR1, OsGID1, SP8, RSV P2 and RSMV M proteins, were cloned by PCR into *pCV1300-cYFP* or *-nYFP* entry vectors under the control of a *CaMV 35S* promoter. These constructs were then transformed into *Agrobacterium tumefaciens* strain GV3101 and transiently co-infiltrated with complementary combinations into approximately 6-week-old *N. benthamiana* leaves. Partners that interact bring their attached cYFP and nYFP physically close, partly rebuilding the fluorescence activity of the YFP vector. The restored YFP fluorescence signal for each combination was detected using a confocal microscope (Leica TCS SP8) 48 h after infiltration. Plasmids and PCR primers for the PCR experiments are listed in Supplementary Table 1.

For competitive BiFC assays, components to be tested were mixed in equal volumes before infiltration into *N. benthamiana*, and MG132 (Sigma) was added at a final concentration of 50 μM to prevent the specific degradation of SLR1. Different combinations were tested in opposite halves of leaves, representative confocal images were taken from at least 15 independent biological repeats, and the relative fluorescence intensity was quantified using ImageJ software (https://imagej.nih.gov/ij/).

### Coimmunoprecipitation (Co-IP)
Co-IP from *N. benthamiana* cells was performed as described previously. The full-length coding sequences of the SLR1, OsGID1, SP8, RSV P2 and RSMV M proteins were amplified by PCR using KOD DNA polymerase (TOYOBO, Osaka, Japan), and then insert into the pCV1300-myc, -flag or -GFP vector by using gene-specific primers (Supplementary Table 1). The combinations to be tested were then infiltrated using *Agrobacterium* into *N. benthamiana* leaves. The samples were harvested at 48 hpi and extracted with IP buffer (25 mM Tris-HCl pH = 7.4, 150 mM NaCl, 1% NP-40, 5% Glycerol, 1 mM DTT and one protease inhibitor cocktail Complete Mini tablet (Roche)), and the mixtures were kept at 4 °C with gentle shaking for 20 min. The supernatant was incubated with 5 μl Flag-trap beads (Genscript, China) for about 2 h at 4 °C and washed three times with cold 1× PBS buffer. After addition of 100 μl loading buffer (1 M Tris-HCl, pH = 6.8, 5% SDS, 0.25% Bromophenol Blue and 25% Glycerol) into beads, the samples were boiled for 5 min and subsequently loaded onto 12% SDS-PAGE gels for immunoblot analysis.

For competitive Co-IP assays, components to be tested were mixed in equal volumes before infiltration using *Agrobacterium* into *N. benthamiana*, and MG132 (Sigma) was added at a final concentration of 50 μM to prevent OsGID1 or SP8/P2 induced degradation of SLR1. Cultures were pelleted to a final $OD_{600}$ of 0.5, increasing amounts of SP8 and P2 following agrobacterium infection with final $OD_{600} = 0.5$ or $OD_{600} = 1.0$, respectively. Samples were harvested at 48 hpi and extracted with IP buffer, then incubated with the corresponding beads

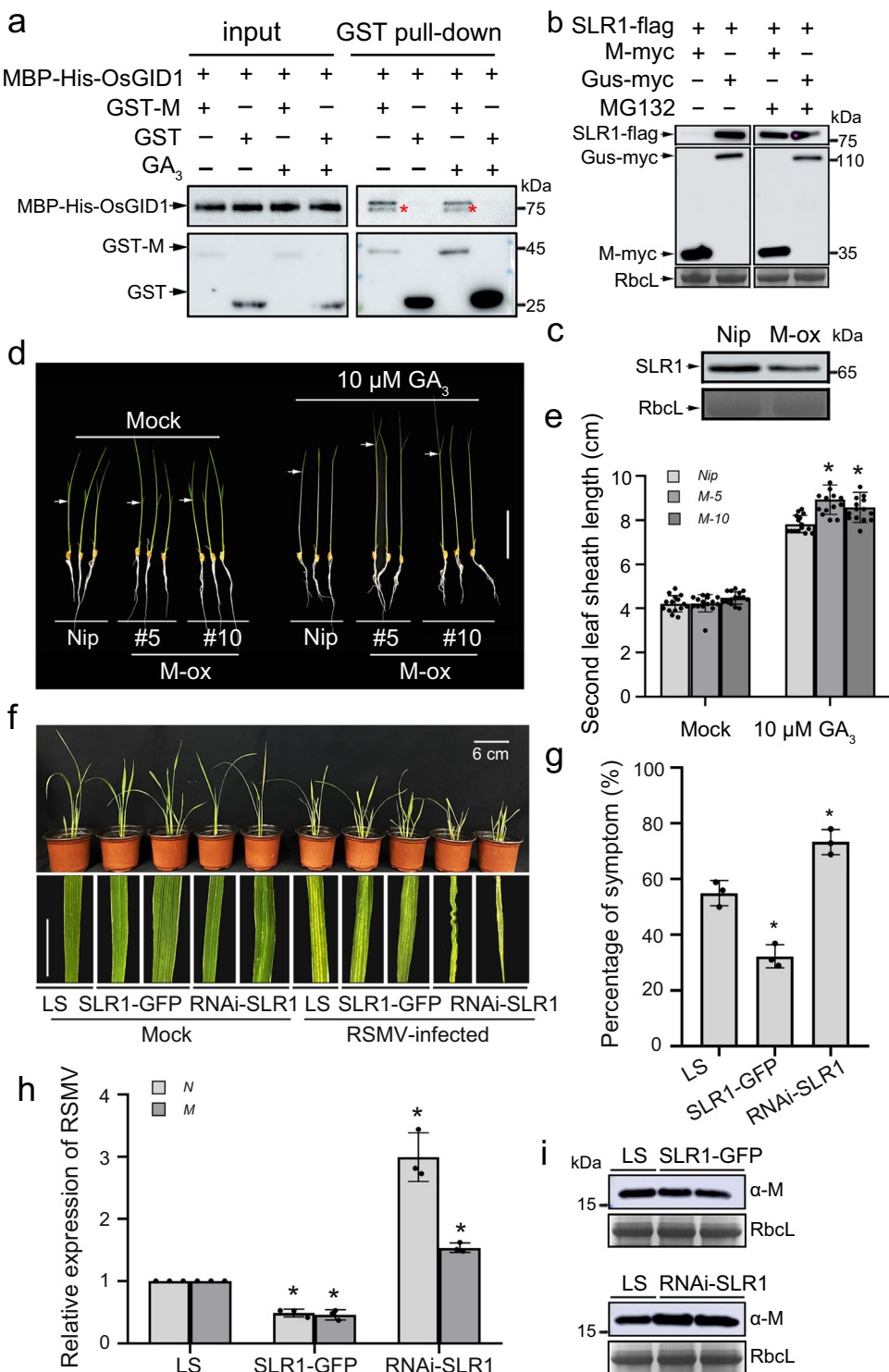

and analyzed by western blot. Generally, protein input was considered as an internal control to monitor the expression level of whole protein in vivo. Protein levels were quantified using ImageJ software(https://imagej.nih.gov/ij/).

**In vitro pull-down assay**

To obtain recombinant proteins in vitro, the cDNAs encoding OsGID1, SP8, P2 and M proteins were inserted into the His-fusion vectors pCold-TF-His and PET32a-MBP-His, respectively. The cDNAs encoding SLR1, SP8, P2 and M proteins were cloned into the pGEX-6P1 vector. Primers used for these constructs are listed in Supplementary Table 1.

These recombinant constructs and the corresponding empty Pcold-TF-His, PET32a-MBP-His and pGEX-6P1 vectors were further transformed into *E. coli* Rosetta (DE3) (Wei Di, China), and protein expression was induced by 10 μM IPTG. The soluble glutathione S-transferase (GST) fusion proteins were directly purified from GSTrap™ FF column (GE), whereas the soluble His-fusion proteins were purified using a Ni-NTA His-trap column (GE).

For pull-down assays, equal amounts of GST-SP8, GST-P2, GST-M or GST with or without 100 μM GA$_3$ were incubated with GST beads (Beaver) at 4 °C for about 1 h, after which PET32a-MBP-His-OsGID1 was added for a further 2 h. The beads were then washed thoroughly,

**Fig. 7 | RSMV M also manipulates SLR1. a** In vitro pull-down assays for analysis of the interaction between OsGID1 with M protein. An equal amount of MBP-His-OsGID1 was incubated with immobilized GST and GST-M, and then the bound proteins were detected by Western blotting using anti-His and anti-GST antibodies. Each experiment was repeated three times with similar results. **b** Effect of viral protein RSMV M on the accumulation of SLR1 in *N. benthamiana* leaves. The co-infiltrated leaves were treated with MG132 (50 μM) or DMSO at 24 hpi and then were harvested for western blotting 24 h later. RbcL was used as a loading control to monitor input protein amount. Each experiment was repeated three times with similar results. **c** Endogenous SLR1 protein levels in wild-type *Nip* or *M-ox* transgenic rice plants. The samples were collected from 7-day-old seedlings for protein extraction, after that, the total proteins were immunoblotted by gel blot with anti-SLR1 antibody. RbcL was used as a loading control to monitor input protein amount. Each experiment was repeated three times with similar results. **d** Phenotypes of *Nip* and *M-ox* seedlings treated with GA₃. Similar germinated seeds were planted in different concentrations of GA₃ (0, 10 μM) containing culture solution for about 7 d, *n* = 3 biologically independent replicates per genotype. All

images were photographed using a digital camera. Scale bar = 4 cm. **e** Second leaf sheath lengths of Nip and M-ox seedlings treated with GA₃. Error bars represent SD, * indicates a significant difference between samples, statistics analysis of data obtained from at least three biological repeats, with 30 plants from each line in every repeat. **f** Symptoms on *LS*, *SLR1-GFP* and *RNAi-SLR1* rice plants following RSMV infection. Values were obtained from *n* = 30 biologically independent plants, *n* = 3 biologically independent replicates per genotype. Photos were taken at 20 dpi. Scale bars = 6 cm. **g** The percentages of *LS*, *SLR1-GFP* and *RNAi*-SLR1 plants with RSMV symptoms. * at the top of columns indicate significant differences (*p* < 0.05) based on Fisher's least significant difference tests. **h** QRT-PCR analysis of the relative expression level of RSMV RNAs (*N* and *M*) in infected *LS*, *SLR1-GFP* and *RNAi-SLR1* rice plants. * at the top of columns indicate significant differences (*p* < 0.05) based on Fisher's least significant difference tests. **i** The accumulation of RSMV M protein in infected *LS*, *SLR1-GFP* and *RNAi-SLR1* rice plants by western blot. RbcL serves as the loading control to monitor input protein amount. Source data including uncropped scans of gels (**a**–**c** and **i**) and *p* values of statistic tests (**e** and **h**) are provided in the Source data file.

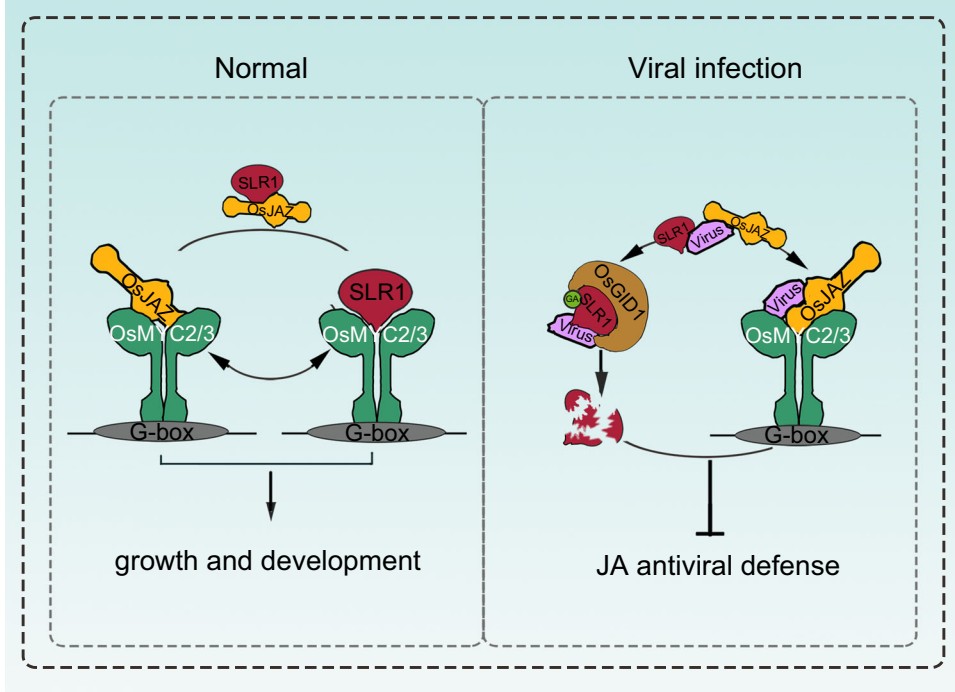

**Fig. 8 | A proposed working model showing the role of the identified viral proteins in disturbing the SLR1-mediated broad-spectrum antiviral network in rice.** Under normal conditions, stabilized SLR1 protein physically competes with OsMYC2/3 for binding to OsJAZ proteins, achieving the dynamic balance that adapts plants to survive in these conditions, thereby facilitating the downstream expression of JA-responsive genes involved in host growth and development. When

challenged by the viruses, SLR1 triggers antiviral JA signaling cascades, but these in turn are counteracted by the independently evolved viral effectors. These viral effectors suppress the JA antiviral response by blocking the association of SLR1 and OsJAZ repressors, and enhancing the affinity of SLR1 with the GA receptor OsGID1, leading to a progressive compromise of host immunity by intercepting the OsMYC-mediated systemic antiviral resistance.

retained beads were resolved by SDS-PAGE and probed with anti-His antibody (1:3000 dilution, ab18184, Abcam).

For competitive pull-down assays, equal amounts of GST-SLR1 and PET32a-MBP-His-OsGID1 were incubated with 100 μM GA₃ followed by 10 or 50 μg pCold-TF-His-SP8 at 4 °C for about 2 h. After washing, proteins retained on the beads were resolved by SDS-PAGE and probed with anti-GST (1:5000 dilution, Cat#A00130, Genscript) or anti-His (1:3000 dilution, ab18184, Abcam) antibody.

**In vivo degradation assay**

For the degradation assay in *N. benthamiana*, individual cultures of SLR1-flag were mixed equally with SP8-myc, RSV P2-myc, RSMV M-myc or Gus-myc. The *N. benthamiana* leaves were then treated with 50 μM MG132 (Sigma) or DMSO at 24 hpi, and the samples were collected at

48 hpi to estimate SLR1 amounts by western blot with anti-flag (1:5000 dilution, Cat#HT201-01, TransGen). RbcL served as a loading control. Each treatment was done in biological triplicate.

**In vitro degradation assay**

To establish an efficient protein purification system, the full-length cDNA fragments of viral proteins SP8, RSV P2 and RSMV M proteins were cloned into the His vector. The recombinant His-SP8, His-P2, His-M and the empty His control were individually transformed into *E. coli* Rosetta-gami (DE3) pLysS, and then purified using His-Trap beads according to the manufacturer's instructions. For in vitro degradation assays, total protein was extracted from 7 day-old wild-type *Nip* seedlings with the IP buffer. Equal amounts of protein supernatant were co-incubated with 20 μg His-SP8, His-P2, His-M or His empty

vector at 30 °C for each experimental group. Samples were taken every 30 min to estimate endogenous SLR1 using anti-SLR1 (1:5000 dilution A18329, ABclonal). RbcL and plant actin antibody served as loading control. Each experiment was repeated three times with similar results.

## Viral inoculation
Viruses were inoculated using their insect vectors as described previously. SRBSDV and RSV were transmitted respectively by the white-backed planthopper (WBPH) and the small brown planthopper (SBPH), while RSMV was transmitted by the leafhopper *Recilia dorsalis*. For virus acquisition, a large number of nymphs of the appropriate vector insects were fed on virus-infected plants for 4 d, and then transferred onto 7-day-old healthy rice seedlings for 12 d to allow the development of viruliferous adults. For virus transmission, the viruliferous adult insects were transferred to the test rice seedlings (approximately three viruliferous insects per seedling) and allowed to feed for 3 d, after which plants were moved into the greenhouse and examined regularly to monitor symptom development.

## RNA extraction and Quantitative RT-PCR
Total RNA from rice leaves was extracted using the TRIzol reagent in accordance with the manufacturer's instructions. 1 μg total RNA was treated with gDNA wiper mix to eliminate genomic DNA and then reverse transcribed to cDNA using HiScript® III qRT Super Mix (Vazyme). There were three biological repeats for each sample. Quantitative RT-PCR (qRT-PCR) was then performed on a Light-Cycler480 II Real-Time PCR System (Roche) using the Hieff qPCR SYBR® Green Master Mix (Yeasen). The qRT-PCR conditions were set as follows: 95 °C for 3 min, 95 °C for 15 s (40 cycles), 60 °C for 15 s and finally 72 °C for 20 s. The relative mRNA expression levels were normalized using the actin gene *OsUBQ5*, and the data ultimately analyzed by the $2^{-\Delta\Delta Ct}$ method.

## GA sensitivity analysis
The elongation of the second leaf sheaths was used as an indicator of GA sensitivity. The tested transgenic rice seedlings (≥20 seeds per line) were germinated under uniform conditions, planted into nutrient solutions containing different concentrations of GA₃ (0, 0.1, 1, 2, 5 and 10 μM) and grown under short day conditions (8 h light, 25 °C/16 h dark, 30 °C). The lengths of the second leaf sheath were measured and recorded after a week, and the phenotypes were photographed using a digital camera. Each experiment was repeated three times with similar results.

## Primary root inhibition assay
To test the JA sensitivity in primary root inhibition assays, seeds of each transgenic genotype (≥20 seeds per line) were germinated under uniform conditions, sprinkled onto floating plates in rice nutrient solution with different concentrations of MeJA (0, 0.1, 1 μM) and maintained under short day conditions (8 h light, 25 °C/16 h dark, 30 °C). Root lengths were measured and photographed after a week, and relative root lengths were used as a measure of JA response.

## Statistical analysis
Differences were analyzed using one-way or two-way ANOVA with Fisher's least significant difference tests. A *p*-value ≤ 0.05 was considered statistically significant. All analyses were performed using ORIGIN 8.0 software.

## Accession numbers
Sequence data from this article can be found in the rice genome annotation project database under the following accession numbers:
*SLR1*, Os03g49990; *OsGID1*, Os05g33730; *OsJAZ3*, Os08g33160; *OsJAZ4*, Os09g23660; *OsJAZ9*, Os03g08310; *OsJAZ12*, Os10g25290; *MYC2*, Os10g42430; *OsMYC3*, Os01g50940.

## Reporting summary
Further information on research design is available in the Nature Portfolio Reporting Summary linked to this article.

## Data availability
The authors declare that all raw data supporting the findings of this study can be found within the paper and its Supplementary Files. Source data are provided with this paper.

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

## Acknowledgements

We thank Dr. Jiayang Li (Institute of Genetics and Developmental Biology, Chinese Academy of Sciences) and Dr. Qian Qian (National Rice Research Institute, Chinese Academy of Agricultural Sciences) for offering the transgenic rice seeds *SLR1-GFP* and *RNAi-SLR1* used in this study, and Prof. Jianxiang Wu (Zhejiang University) for providing viral proteins antibody. We are indebted to Prof. Mike Adams for critically reading and improving the manuscript. This work was funded by the National Key Research and Development Plan of China (2021YFD1400500), National Natural Science Foundation of China (32022072, 32172416, 32100103), Zhejiang Provincial Natural Science Foundation (LZ22C140001, LR22C140002).

## Author contributions

L.L., J.C. and Z.S. designed the experiments, L.L. performed most experiments with assistance from H.Z., Z.Y., C.W., S.L., C.C., T.Y., Z.W. and Y.L., L.L., J.C. and Z.S. analyzed and discussed the results; L.L. and Z.S. wrote the manuscript.

## Competing interests

The authors declare no competing interests.
