## [Peer Review File · Nature Communications]

Independently evolved viral effectors convergently suppress
DELLA protein SLR1-mediated broad-spectrum antiviral
immunity in riceReviewer #1 (Remarks to the Author):

The authors Li et al in their paper 'Independently evolved viral effectors convergently suppress DELLA protein SLR1-mediated broad spectrum antiviral immunity in rice' nicely describe the interaction of viral effectors from diverse viruses with the main rice DELLA protein SLR1 and the combined effect on interaction with JAZ, MYC and GID1 proteins. They also evaluate separately the effect of these interaction on growth through the GA pathway or through JA pathway. Initial experiments were performed with 4 viral proteins, majority with three and some with two. I agree with the general message given although in some of the experiments the data do not totally support the claims they make.

-the effect of GA on the growth of SP8-ox plants is not so strong as for P2-ox plants as the effect on growth was observed in only one line. The authors should alleviate their claims for the effect on growth. Or at least give some explanation. For sure these are plants that constitutively expressed viral effectors and the plants most probably adapted to their effects on GA signaling. Thus the phenotypes are not very strong

-authors should take care to only imply that this system is true for rice (like they did appropriately in the title but not always in abstract and discussion) as this is not necessary true for all plants. If they would like to claim that they should also test some dicot plant-virus interactions

- Some information on interference of viruses with GA signaling is also known for dicot plants. Please check the literature and properly include information in discussion.

- the authors have nicely shown the consequences of viral manipulation of both GA and JA signaling on growth. But the link to the immunity and consequent broad resistance is missing. Can you give at least some implication how both help to reduce symptoms and viral multiplication in the plant?

-interaction of viral effectors with GID1 was shown only with CoIP. As viral effectors interact also with DELLA which are naturally expressed in *N.benthamiana* the results can be the result of pull down of the complex. Additional Y2H experiments should be performed to confirm direct interaction. Or alternatively change the text to indicate the possibility of indirect interaction

-Fig 1 and 2 results of CoIP experiments should have the indications of position of proteins of interest similarly as you did in CoIP experiments for Fig 3

-according to MIQE standards the authors should provide the information on efficiency of amplification for the amplicons they have designed de-novo for qPCR. If the amplicon was not designed in this study information on the original publication should be given

- why did you put data on M protein into separate chapter – they could be explained in parallel with SP8 and P2

Reviewer #2 (Remarks to the Author):

In this manuscript the authors have characterized the interaction of rice virus proteins with SLR1, and have also explored the effect of this interaction on the virus infection as well as the effect on plant disease. The manuscript is a mixture of experiments that can be evaluated objectively, coupled with other experiments that are more subjective in nature. On the objective experiment side, the authors have shown that virus proteins SP8 and P2 interact with SLR1 and OSGID1. The subjective experiments involve the degradation of SLR1 by SP8 and P2, the role of SP8 and P2 in promoting the interaction between OSGID1 with SLR1, and the role of SLR1 conferring virus resistance in rice. Each of these subjective experiments is discussed below.

1. Degradation assays. In Figs. 2c and d the authors purify His-SP8 and GST-P2 and add these proteins to total proteins extracted from 7 day-old rice seedlings. The authors claim that the addition of either SP8 or P2 accelerates the degradation of SLR1 in a time course experiment. I did not find these results convincing. In looking at Fig. 2c, the addition of His-SP8 is hypothesized to accelerate the degradation SLR1. However, SLR1 appears to degrade over time even in the absence of SP8, so it is not clear to me how much SP8 is accelerating the process. The authors state that this experiment is representative of the three conducted, but it should be possible to quantitate the percent degradation and subject it to a statistical analysis to show that the degradation is significantly greater in the presence of SP8.

Instead of presenting multiple timepoints, the authors could quantify SLR1 at the start and end of the assay. Also, I am not sure that RbcL is the appropriate loading control, given that it is present at a much higher amount than SLR1. It is not clear from the Materials and Methods if RbcL was detected by western or Coomassie stain (I assume Coomassie). A better control would be some other plant protein expressed at a level comparable to SLR1, which would presumably illustrate that any degradation induced by SP8 or P2 is specific to SLR1.

As second issue with the degradation assays occurs with Fig. 2e. In this figure the authors show that the level of SLR1 is lower in transgenic plants that express SP8 or P2 than in Nippon. They need to also present evidence showing SLR1 mRNA levels are unaffected in the transgenic plants relative to the controls to prove that any effect is due to protein degradation.

2. Effect of virus proteins on the interactions of SLR1 and OsGID1. The authors present experiments in Fig. 3c and d suggesting that increasing amounts of SP8 or R2 will increase the interaction of SLR1 with OsGID1. I am not convinced with these figures. The authors do not explain how SP8 and P2 amounts were increased in these experiments. Presumably, the proteins were introduced through agroinfiltration, but no conditions such as OD600 are included in this paper. Furthermore, the Westerns for SP8 do not show conclusively that there are progressive increases in the two lanes expressing SP8 or P2. More importantly, the authors need to quantify the amounts of SLR1 and OsGID1 in each of the lanes to convince me that SP8 or P2 have any effect on the interaction. What is the increase in recovery of these proteins in the presence of SP8?

In Fig. 3e and f, the authors present BiFC experimental results obtained with a confocal microscope and claim that the addition of SP8 or R2 enhances the interaction between OsGID1 and SLR1. However, not enough information has been given in the narrative, figure legend or Materials and Methods to determine how the images were selected for analysis. Expression levels of proteins in *N. benthamiana* will vary from one leaf to another and also will vary within the zone of infiltration. Consequently, the authors would need to compare expression levels of two treatments in opposite half leaves. However, even this would fall short, because attempts to choose a "representative" level of expression may be completely subjective. They authors need to make their selections for expression levels in an unbiased manner, and clearly explain how this was accomplished. Alternatively, the authors might try an approach similar to the one they used in Supplemental Fig. 5.

3. Influence of SLR1 expression on virus symptom development in Figs. 4a, 5a, and 7e. To determine the effect of SLR1 modification on viral symptom development, the authors score their plants as "healthy without symptoms (N), typical yellow stripes (I) and curling or death (II) of young leaves. However, in examining Fig. 4a and 5a, the overwhelming symptom that jumps out at me is stunting; there does not appear to be any difference in the degree of stunting in any of the three types of plants (LS, SLR1-GFP, RNAi-SLR1) inoculated with either RSV or SRBSDV. The heading for Figs. 4 and 5 is that SLR1 confers resistance to these viruses, yet both viruses stunt the growth of plants regardless of SLR1 status. The authors appear to focus on one type of symptom (yellow stripes and necrosis), while ignoring the effect of virus infection on the overall size of plants. Any effect on symptom type, yellow streaks, or necrosis, needs to be re-evaluated with the inclusion of the apparent stunting phenotype.

The authors also present evidence that alterations in SLR1 affect expression of RSV and SRBSDV, as assessed through quantitative RT-PCR. These values are valid. I agree that overexpression of SLR1 has a negative effect on viral RNA expression, and that silencing of SLR1 leads to increases in viral RNA expression. However, I am not convinced that this translates to milder symptoms, based on the figures presented in this manuscript.

4. Influence of viral proteins on the interaction of SLRa and Jaz proteins. The criticism for Fig 3c also applies to Fig. 6a-d. Fig. 6d is especially problematic, because SLR1 is absent from lanes 3 and 4 of the Flag-IP blot. The interpretation of this is that OsMYC3-flag level is reduced in the absence of SLR1, or an alternate interpretation is that we are looking at normal variation in protein levels in a Co-IP.

Other comments.

Fig. 1, and associated narrative for BiFC assay (lines 141-146). The authors negative control shows that no YFP signal is observed for the combination SLR1-cYFP/Gus-nYFP. They also should include the negative controls for SP8-nYFP, P2-nYFP, and M-nYFP.

Fig.1 and its legend, lines 746-756. Panels 1c and 1e have red stars next to specific bands. Explain the red stars in the figure legend. Also, should bands in 1d also be labeled with red stars? Be sure throughout the paper to include an explanation for the red stars adjacent to bands in the figures.

Lines 150-153. The authors describe Co-IPs between SLR1 and SP8, P2, and M. In subsequent Co-IPs, the authors mention that the necessity of pre-treatment with MG132 to stabilize SLR1. Was MG132 also used in Fig. 1c-e?

Lines 153-155. The authors conclude that SLR1 is a conserved interaction partner via its GRAS domain with the viral proteins SP8/P2/M. The evidence for interaction specifically with the GRAS domain only is observed through the yeast two-hybrid screen. For such a conclusive statement, it would be good if the interaction with the GRAS domain was confirmed with either the BiFC assay or co-IP assay.

Lines 313-317, and supplemental Fig. 5. Authors state, "... increasing the quantity of SLR1 progressively decreased the YFP fluorescence signals of the OsJAZ9-cYFP/OsMYC3-nYFP combinations." Supplemental Fig. 5 presents only a single agroinfiltration of SLR1. In this type of assay, it is important to provide the OD600 for the agrobacterium isolates used for agroinfiltration. Finally, to evaluate the effect of SLR1 on OsJAZ9-cYFP/OsMYC3-nYFP, the two treatments need to be conducted on half leaves of the same *N. benthamiana* leaf, and multiple leaves should be evaluated in this manner.

Lines 325-327. Supplemental Fig. 7 shows results of a Y2H screen for evidence of interaction of GRAS domain with OsJaz proteins. As with Fig. 1, are Y2H interactions always representative of the interactions in plants?

Reviewer #3 (Remarks to the Author):

In this study entitled "Independently evolved viral effectors convergently suppress DELLA protein SLR1-mediated broad-spectrum antiviral immunity in rice", the authors found three rice viruses encoded proteins (SRBSDV SP8, RSV P2 and RSMV M) interacted with the general target protein SLR1 in vitro and in vivo, and trigger rapid degradation of SLR1 by promoting the interaction of GA receptor OsGID1 with SLR1, and also diminish the ability of SLR1 to activate JA signaling by disassociating SLR1 from the OsJAZ-OsMYC2/3 complex, leading to repressing SLR1-mediated broad-spectrum antiviral defense to viral infection. The findings are of significance to the pathogen-host interaction field and related fields. This is an original work, the data support the conclusions and claims, and the methodology is sound and the work meet the expected standards in the field. Therefore, the work is suitable for publication in the journal after minor revisions.

My comments on minor revisions:

1. English language should be extensively checked, pay more attention to verb tense agreement.
2. Line 383, please delete one "that".
3. Most references cited are not unified in format, in particular authors' name.

Reviewer #4 (Remarks to the Author):

In this study, the authors reported that the molecular mechanism of infection by viruses is related to GA signaling. First, inspired by previous studies showing that viral proteins function as key

factors in JA and auxin signaling, the authors found that SLR1, a key factor in GA signaling, binds to viral proteins and promotes degradation of SLR1. Using rice SLR1 knockdown mutants and overexpressors, as well as rice plants overexpressing viral proteins, they showed that SLR1 functions to suppress viral infection and that its action is inhibited by viral proteins. In addition, based on their previous results, the authors investigated the effects of SLR1 and viral proteins on the action of JA signaling key factors JAZs and MYC2/3 using molecular biological and physiological studies to clarify the regulatory mechanism of GA-JA signaling in rice. Finally, they concluded that viral proteins hijack this mechanism to spread the infection.

The authors succeeded to show that the viral proteins bind to and regulates the degradation of SLR1, which could be the first direct observations of a molecular mechanism for direct crosstalk between pathogen infection and GA signaling. Thus, I consider that the findings could be potentially evaluated as novel in terms of plant molecular physiology and pathology. However, some of results presented here differ from previously reported observations, particularly on GA signaling, and they need to carefully check the reliability of these results. Some of the problems are pointed out below.

Fig. 2e shows that overexpression of SP8 or P2 greatly reduced the amount of SLR1 in the plant. Despite this, the height of these plants in panel f/h was similar to that of the control plants (Nip), which is completely different from the previous observation. The authors also noticed such discrepancy and gave the following excuse; No differences in plant height were observed between these plants, perhaps partly because there are only small amounts of endogenous SLR1 in Nip background rice plants, and although SLR1 was degraded in the transgenic plants, the differences were insufficient to cause phenotypic effects. However, this explanation is not consistent with the results of Fig. 2e. In fact, SLR1 in the Nip background is clearly observed in Fig. 2e. In addition, overexpression of SP8 or P2 greatly reduces SLR1 in plants, so the differences were not insufficient. In this paper, I often see this kind of interpretation that favors the authors' hypothesis (in one case, the GA signal is de-repressed as a result of the degradation of SLR1, and in another case, SLR1 is degraded but the GA signal remains unchanged). Such an attitude undermines the credibility of this paper.

Fig.2c and 2d also have a major problem. In this experiment, the authors used "total protein extracted from seedlings" to observe the degradation of SLR1 by GA3, but the degradation of SLR1 is not reproducible. In fact, its degradation pattern of "+His+GA3" and "+GST+GA3" is not the same, but the results of both are clearly different. Furthermore, in "+His+SP8+GA3", SLR1 seems to be degraded very slowly, while in "+GST+P2+GA3", SLR1 is partially degraded in the first 60 minutes and no further degradation is observed. Based on these results, they should assume that they cannot reproducibly observe the serial change of SLR1 by GA in this experimental system. The results in Fig. 2g and 2i are also questionable. This experiment was conducted to investigate the effect of GA3 concentration on the elongation of rice seedling. In a previous experiment, the GA3 concentration on the elongation of the second leaf sheath length of rice seedling was reported to be 10⁻¹³ to 10⁻¹⁰ M (e.g. *Planta* 205(2):145, 1998). Since the GA concentration used in this experiment is much higher than that, they cannot discuss effect of GA concentration. Furthermore, the fact that the leaf sheath length of SP8-ox and P2-ox is longer than that of Nip even at very high concentrations of GA (even though GA signaling is fully saturated) suggests that this difference is due to something other than GA signaling.

In Fig. 3, the authors examine the formation of GID1-SLR1-SP8/P2 in transiently infiltrated tobacco leaves. They showed that the SP8/P2 protein binds to GID1 and SLR1 individually (presumably in the absence of GA), but they did not say whether GA is required for this GID1-SLR1-SP8/P2 formation. What effect the involvement of SP8/P2 in OsGID1-SLR1 interaction has on the degradation of SLR1 (e.g., responsiveness to GA and rate of SLR1 degradation) is very important for this paper. However, this paper does not mention this at all. This should perform some new experiments using in vivo system (not in vitro) to answer these questions.

Fig. 4 also contains a similar problem. Why is Ri-SLR1 not growing taller? Is the amount of SLR1 really changing under these conditions? If so, does the change in the amount of SLR1 affect GA signaling? As expected, the SLR1-GFP plants show dwarfism. Then, how much accumulation of SLR1 occurs in these plants to suppress its plant height? In Fig. 4, the authors observed only changes in viral proteins and virus symptoms, but paid no attention to the quantitative changes in SLR1 even though they want to discuss the relationship between SLR1 and viral propagation. About the result of Fig. 7b, they discussed that "endogenous SLR1 was more rapidly degraded". Of course, they cannot make such discussion on the kinetics of SLR1 degradation by using this result. The discussion on the kinetics of SLR1 degradation and that of GA-dose dependence should be

very important for this paper, because when the virus components are really involved in the GA signaling via SLR1, the events caused by viral proteins should mimic the stereotype GA-response. The authors should pay attention to the plant physiological aspects of GA-induced phenomena and re-examine the physiological phenomena during viral infection, which is the subject in this paper. The presentation and explanation of Fig. 8 is inappropriate and is not a good summary of this paper. When SLR1 binds to OsMYC2/3, it facilitates JA signaling resulting in inhibiting the rice growth/development. This figure does not show that state. On the other hand, in the presence of viral infection (viral proteins), SLR1 degradation is accelerated and binding to OsMYC2/3 is inhibited, resulting in accelerated binding of OsJAZ and viral proteins to OsMYC2/3 (right side of viral infection in the figure). In this state, JA signaling should be reduced, and plant growth should be free from its inhibitory state and should be growth-promoting. However, in the figure, this state is represented by a dwarfed plant with a T-formed line indicating inhibition. This dwarfed plant (probably) indicates a state in which viral resistance is weakened due to reduced JA signaling and plant growth is inhibited as a result of increased viral infection, but it is clearly inappropriate to represent this as a T-formed line on the plant.

Minor comments

About Fig. S5, the text mentioned that they quantified the YFP signals by OsJAZ9-OsMYC3 complex under the presence/absence of SLR1. However, there is no information on how to quantify the signals. They need to describe the details about this experiment.

Response to Reviewers' Comments

We are very grateful to the reviewers for their valuable comments and suggestions on the manuscript. We have carefully taken these into consideration in preparing this revision and have performed the additional experiments suggested. We believe that the revised manuscript has addressed the reviewers' concerns and has resulted in a better paper. Comments received are shown in black below, with our response in red font. The following are our point-by-point answers to the reviewers' questions and comments.

Reviewer #1 (Remarks to the Author):

The authors Li et al in their paper 'Independently evolved viral effectors convergently suppress DELLA protein SLR1-mediated broad spectrum antiviral immunity in rice' nicely describe the interaction of viral effectors from diverse viruses with the main rice DELLA protein SLR1 and the combined effect on interaction with JAZ, MYC and GID1 proteins. They also evaluate separately the effect of these interaction on growth through the GA pathway or through JA pathway. Initial experiments were performed with 4 viral proteins, majority with three and some with two. I agree with the general message given although in some of the experiments the data do not totally support the claims they make.

-the effect of GA on the growth of SP8-ox plants is not so strong as for P2-ox plants as the effect on growth was observed in only one line. The authors should alleviate their claims for the effect on growth. Or at least give some explanation. For sure these are plants that constitutively expressed viral effectors and the plants most probably adapted to their effects on GA signaling. Thus the phenotypes are not very strong.

Response: Thank you for your valuable comments. We have re-analyzed the effect of GA₃ on the growth of SP8-ox plants (Fig. 2e-h). Additionally, we

carried out qRT-PCR to monitor the relative expression level of *SP8* in different transgenic lines, and we found that the expression of *SP8* in line #13 was obviously lower than that in the lines #24 and #26 (see below Figure 1). Hence, we have chosen lines #24 and #26 for GA₃ sensitivity assays and have shown that both these lines are clearly more sensitive to GA₃ treatment. We think that these phenotypic effects may be due to the differential expression of *SP8* in transgenic rice plants.

Figure 1. GA₃ sensitivity of plants constitutively expressing *SP8*.

-authors should take care to only imply that this system is true for rice (like they

did appropriately in the title but not always in abstract and discussion) as this is not necessary true for all plants. If they would like to claim that they should also test some dicot plant-virus interactions

Response: Thank for your valuable advice. We have modified this in our revised manuscript (Line 37, Line 473, Lines 508-509).

- Some information on interference of viruses with GA signaling is also known for dicot plants. Please check the literature and properly include information in discussion.

Response: We have now added this and cited the relevant literature (Lines 435-446, References 46 to 47).

- the authors have nicely shown the consequences of viral manipulation of both GA and JA signaling on growth. But the link to the immunity and consequent broad resistance is missing. Can you give at least some implication how both help to reduce symptoms and viral multiplication in the plant?

Response: We really thank the reviewer for raising this meaningful issue, and we have added some discussion about this in the revised manuscript (Lines 488-503). Plants have a complicated defense signaling network, including RNA silencing, resistance gene (*R* gene)-mediated defense and activation of phytohormone signaling. Increasing evidence shows that jasmonic acid (JA) is a crucial hormone in plant antiviral immunity (He et al., 2017, Li et al., 2021). One of the outcomes of JA recognition and the subsequent signaling cascades is the expression of pathogenesis-related (*PR*) genes, thus inducing the initial immunity response to viral infection (Yang et al., 2013). Recently, a strong link between JA-mediated signaling and RNA silencing was demonstrated, and accumulating JA promotes rice antiviral defense through inducing *AGO18* expression (Yang et al., 2020). Our study provides strong evidence for a key role of SLR1 in JA-mediated broad-spectrum defense against distinct virus

infection. However, the underlying molecular regulatory mechanisms, especially the way in which SLR1 and JA signaling reduces symptoms and viral multiplication, are still incompletely understood. We here identified that SLR1 activates the JA-antiviral response to reduce symptoms and viral multiplication by competitively binding with the key OsMYC2/3 transcription factors. This will affect the complicated downstream antiviral regulatory network, perhaps by upregulating the expression of *PR* genes, by degrading viral mRNAs due to crosstalk with RNA interference or by other as yet unknown mechanisms.

References:

1. He Y, et al. Jasmonic acid-mediated defense suppresses brassinosteroid-mediated susceptibility to Rice black streaked dwarf virus infection in rice. *New Phytol* **214**, (2017)
2. Yang D, et al. Roles of Plant Hormones and Their Interplay in Rice Immunity. *Molecular Plant* **6**, (2013).
3. Li L, et al. A class of independently evolved transcriptional repressors in plant RNA viruses facilitates viral infection and vector feeding. *Proc Natl Acad Sci USA* **118**, (2021).
4. Yang Z, et al. Jasmonate Signaling Enhances RNA Silencing and Antiviral Defense in Rice. *Cell Host & Microbe* **28**, (2020).

-interaction of viral effectors with GID1 was shown only with Co-IP. As viral effectors interact also with DELLA which are naturally expressed in *N.benthamiana* the results can be the result of pull down of the complex. Additional Y2H experiments should be performed to confirm direct interaction. Or alternatively change the text to indicate the possibility of indirect interaction

Response: We have done the yeast two-hybrid assays suggested (Supplementary Fig. 4) and also *in vitro* pull-down assays (Fig. 3c-d and Fig. 7a, see below Figure 2). Overall, these results comprehensively confirmed that SP8, RSV P2 and RSMV M proteins directly interact with OsGID1 independently of GA.

Figure 2. Direct interactions between viral proteins and OsGID1 receptor performed by *in vitro* pull-down assays.

-Fig 1 and 2 results of Co-IP experiments should have the indications of position of proteins of interest similarly as you did in Co-IP experiments for Fig 3.

Response: Thanks: we have now done this.

-according to MIQE standards the authors should provide the information on efficiency of amplification for the amplicons they have designed de-novo for qPCR. If the amplicon was not designed in this study information on the original publication should be given.

Response: The primers for *SLR1* used in qRT-PCR was that described by Liao and colleagues (Liao et al., 2019). The primers used to detect SRBSDV (*S2*, *S4*, *S6*), RSV (*CP*) and RSMV (*M* and *N*) were thoroughly described in Zhang et al. and Li et al. (Zhang et al., 2020, Li et al., 2021).

References:

1. Liao Z, et al. SLR1 inhibits MOC1 degradation to coordinate tiller number and plant height in rice. *Nature Communications* **10**, (2019).
2. Zhang H, et al. Distinct modes of manipulation of rice auxin response factor OsARF17 by different plant RNA viruses for infection. *Proc Natl Acad Sci USA* **117**, (2020).
3. Li L, et al. A class of independently evolved transcriptional repressors in plant RNA viruses facilitates viral infection and vector feeding. *Proc Natl Acad Sci USA* **118**, (2021).

- why did you put data on M protein into separate chapter – they could be explained in parallel with SP8 and P2.

Response: RSMV is newly described and *M-ox* transgenic plants were recently constructed. Due to time constraints, we did not obtain *SLR1-GFP / M-ox* hybrid plants and so we put data on M protein into the last chapter. Despite the lack of genetic hybrid experiments, protein degradation assays (Fig. 7b and 7c, Supplementary Fig. 12), GA₃ sensitivity (Fig. 7d and 7e) and viral inoculation assays (Fig. 7f-i) sufficiently confirmed that RSMV M protein represses SLR1-mediated antiviral defense by promoting the degradation of SLR1.

Reviewer #2 (Remarks to the Author):

In this manuscript the authors have characterized the interaction of rice virus proteins with SLR1, and have also explored the effect of this interaction on the virus infection as well as the effect on plant disease. The manuscript is a mixture of experiments that can be evaluated objectively, coupled with other experiments that are more subjective in nature. On the objective experiment side, the authors have shown that virus proteins SP8 and P2 interact with SLR1 and OSGID1. The subjective experiments involve the degradation of SLR1 by SP8 and P2, the role of SP8 and P2 in promoting the interaction between OSGID1 with SLR1, and the role of SLR1 conferring virus resistance in rice. Each of these subjective experiments is discussed below.

Response: We thank reviewer #2 and have tried to incorporate further experiments in the revised manuscript as suggested, which have improved the manuscript significantly (see below).

1. Degradation assays. In Figs. 2c and d the authors purify His-SP8 and GST-P2 and add these proteins to total proteins extracted from 7 day-old rice seedlings. The authors claim that the addition of either SP8 or P2 accelerates the degradation of SLR1 in a time course experiment. I did not find these results convincing. In looking at Fig. 2c, the addition of His-SP8 is hypothesized to accelerate the degradation SLR1. However, SLR1 appears to degrade over time even in the absence of SP8, so it is not clear to me how much SP8 is accelerating the process. The authors state that this experiment is representative of the three conducted, but it should be possible to quantitate the percent degradation and subject it to a statistical analysis to show that the degradation is significantly greater in the presence of SP8. Instead of presenting multiple timepoints, the authors could quantify SLR1 at the start and end of the assay.

Response: Thanks, we agree with this constructive comment. We think that

the degradation of SLR1 over time even in the absence of SP8 is probably because endogenous OsGID1 and gibberellins (GAs) were present in the cell-free protein degradation system. In the revised manuscript, we present new protein degradation assays at the same time point with SP8 and P2. Compared with the control, endogenous SLR1 degraded more quickly when co-incubated with equal amounts of recombinant SP8 or P2 proteins (Fig. 2c and d). Additionally, as the reviewer suggested, we have subjected the pictures of western blots to Image J software for a statistical analysis (see below Figure 3).

Figure 3. SP8 and RSV P2 provoke rapid degradation of SLR1 in cell-free system.

Also, I am not sure that RbcL is the appropriate loading control, given that it is present at a much higher amount than SLR1. It is not clear from the Materials and Methods if RbcL was detected by western or Coomassie stain (I assume Coomassie). A better control would be some other plant protein expressed at a level comparable to SLR1, which would presumably illustrate that any degradation induced by SP8 or P2 is specific to SLR1.

Response: Thanks for your advice. We did use the well-established method with RbcL and Coomassie stain. However, we agree that a better control would be some other plant protein expressed at a level comparable to SLR1. Thus, we used a plant actin and appropriate antibody in SLR1 protein degradation assays when illustrating the specific degradation induced by distinct viral

proteins (Fig. 2c, d and Supplementary Fig. 12).

As second issue with the degradation assays occurs with Fig. 2e. In this figure the authors show that the level of SLR1 is lower in transgenic plants that express SP8 or P2 than in Nippon. They need to also present evidence showing SLR1 mRNA levels are unaffected in the transgenic plants relative to the controls to prove that any effect is due to protein degradation.

Response: We thank the reviewer for pointing out this relevant issue that we had previously overlooked. We have now analyzed transgene expression by qRT-PCR and this has been added as Supplemental Fig. 3. This showed that there were no significant differences in *SLR1* expression between *SP8-ox* or *P2-ox* lines and the control *Nip* plants. See below Figure 4.

Figure 4. Effect of viral proteins on SLR1 protein and mRNA level.

2. Effect of virus proteins on the interactions of SLR1 and OsGID1. The authors present experiments in Fig. 3c and d suggesting that increasing

amounts of SP8 or R2 will increase the interaction of SLR1 with OsGID1. I am not convinced with these figures. The authors do not explain how SP8 and P2 amounts were increased in these experiments. Presumably, the proteins were introduced through agroinfiltration, but no conditions such as OD600 are included in this paper. Furthermore, the Westerns for SP8 do not show conclusively that there are progressive increases in the two lanes expressing SP8 or P2. More importantly, the authors need to quantify the amounts of SLR1 and OsGID1 in each of the lanes to convince me that SP8 or P2 have any effect on the interaction. What is the increase in recovery of these proteins in the presence of SP8?

Response: We thank the reviewer for raising this important issue. We have added the explanation in the Materials and Methods (Lines 571-581, Lines 938-941, Lines 1016-1018, Lines 1024-1026 and Lines 1032-1035). We have also re-done the Co-IP assays to address the concerns raised and added *in vitro* competitive pull-down assays (Fig.3e, g and h). All the results support our conclusion that viral proteins SP8 and P2 consistently increased the interaction of SLR1 with OsGID1 (Lines 243-251 and Lines 601-605). See below Figure 5.

Figure 5. SP8 and RSV P2 promote interaction of OsGID1 with SLR1 in *in vitro* pull-down protein competition assays.

In Fig. 3e and f, the authors present BiFC experimental results obtained with a confocal microscope and claim that the addition of SP8 or R2 enhances the interaction between OsGID1 and SLR1. However, not enough information has been given in the narrative, figure legend or Materials and Methods to determine how the images were selected for analysis. Expression levels of proteins in *N. benthamiana* will vary from one leaf to another and also will vary within the zone of infiltration. Consequently, the authors would need to compare expression levels of two treatments in opposite half leaves. However, even this would fall short, because attempts to choose a “representative” level of expression may be completely subjective. They authors need to make their selections for expression levels in an unbiased manner, and clearly explain how this was accomplished. Alternatively, the authors might try an approach similar to the one they used in Supplemental Fig. 5.

Response: The use of fluorescent intensity in this context is very common in the literature and we entirely agree that the images and analysis need to be handled in an unbiased manner. Here, SLR1-cYFP and OsGID1-nYFP were co-expressed with SP8-myc (P2-myc) vs. Gus-myc in opposite halves of *N. benthamiana* leaves with repeats and then both halves were examined by confocal microscopy and then quantified and analyzed using the software Image J (Supplemental Fig. 5). We have added a more detailed description about how images were selected for analysis in the Materials and Methods (Lines 547-553).

In the revised manuscript, we have added *in vitro* competitive pull-down assays, which gave similar results demonstrating that the interaction between His-OsGID1 and GST-SLR1 was enhanced by an increased amount of His-SP8 or His-P2 in the presence of GA₃ (Figure 3g and h). Together, these data support our previous conclusion that viral proteins SP8 and RSV P2 could indeed promote the biological association of OsGID1 with SLR1 in *planta* (Lines 243-251 and Lines 601-605).

3. Influence of SLR1 expression on virus symptom development in Figs. 4a, 5a, and 7e. To determine the effect of SLR1 modification on viral symptom development, the authors score their plants as “healthy without symptoms (N), typical yellow stripes (I) and curling or death (II) of young leaves. However, in examining Fig. 4a and 5a, the overwhelming symptom that jumps out at me is stunting; there does not appear to be any difference in the degree of stunting in any of the three types of plants (LS, SLR1-GFP, RNAi-SLR1) inoculated with either RSV or SRBSDV. The heading for Figs. 4 and 5 is that SLR1 confers resistance to these viruses, yet both viruses stunt the growth of plants regardless of SLR1 status. The authors appear to focus on one type of symptom (yellow stripes and necrosis), while ignoring the effect of virus infection on the overall size of plants. Any effect on symptom type, yellow streaks, or necrosis, needs to be re-evaluated with the inclusion of the apparent stunting phenotype. The authors also present evidence that alterations in SLR1 affect expression of RSV and SRBSDV, as assessed through quantitative RT-PCR. These values are valid. I agree that overexpression of SLR1 has a negative effect on viral RNA expression, and that silencing of SLR1 leads to increases in viral RNA expression. However, I am not convinced that this translates to milder symptoms, based on the figures presented in this manuscript.

Response: We apologize that our description has caused some confusion. Actually, only RSV could be scored as “healthy without symptoms (N), typical yellow stripes (I) and curling or death (II) of young leaves”. To our knowledge, diverse RNA viruses are widely distributed and cause holistic dwarfism, darkened leaves or stripes, chlorosis, and necrosis of rice plants, resulting in serious yield losses, but it is specifically RSV that causes chlorosis, weakness and necrosis in emerging leaves, and as a result plant growth is stunted (Yang et al., 2020) (See below Figure 6a). However, RBSDV and SRBSDV mainly causing severe growth abnormalities and particularly severe dwarfism (Zhang et al., 2019) (See below Figure 6b). We have provided more details in the

Introduction (Line 58, Lines 67-69 and Lines 73-75).

Figure 6. Symptom of RBSVDV- and RSV-infected rice plants

In Fig. 4a (lower panel), upon challenging with RSV, the areas of typical yellow stripes and curling or death of the young leaves represent the degree of disease symptoms. According to the severity of symptoms on leaves, we clearly found that *RNAi-SLR1* plants exhibited severe curling or death of the young leaves, whereas *SLR1-GFP* plants only had milder virus symptoms with discontinuous yellow stripes and necrotic streaks on the leaves. Therefore, combined with the data of viral accumulation, we are convinced that SLR1 plays vital roles in rice antiviral defense against RSV infection. However, in Fig. 5a, upon challenging with SRBSVDV, we focused on the viral-induced dwarfism phenotype, which is widely accepted as a good indicator of SRBSVDV infection. In contrast to RSV, no chlorosis, weakness and necrosis was found in SRBSVDV-infected leaves. We therefore measured the severity of SRBSVDV symptoms based on plant height ($SLR1-GFP > LS > RNAi-SLR1$), indicating that SRBSVDV-induced dwarfism was slight in rice plants overexpressing *SLR1-GFP*, despite the fact that *SLR1-GFP* plants appeared to be shorter than *LS* and *RNAi-SLR1* when mock-inoculated. Results from qRT-PCR and Western blot were consistent with the dwarfism and confirm that SLR1 plays

essential roles in rice antiviral defense against SRBSDV infection.

References:

1. Zhang H, *et al.* Suppression of auxin signalling promotes rice susceptibility to Rice black streaked dwarf virus infection. *Molecular Plant Pathology* **20**, 1093-1104 (2019).
2. Yang Z, *et al.* Jasmonate Signaling Enhances RNA Silencing and Antiviral Defense in Rice. *Cell Host & Microbe* **28**, 89-103.e108 (2020).

4. Influence of viral proteins on the interaction of SLRa and Jaz proteins. The criticism for Fig 3c also applies to Fig. 6a-d. Fig. 6d is especially problematic, because SLR1 is absent from lanes 3 and 4 of the Flag-IP blot. The interpretation of this is that OsMYC3-flag level is reduced in the absence of SLR1, or an alternate interpretation is that we are looking at normal variation in protein levels in a Co-IP.

Response: Thanks for the reviewer's helpful comment. In our revised manuscript, to solve the reviewers' concern, we re-performed competitive Co-IP assays in Fig. 6d (See below Figure 7).

Figure 7. P2 disturbed the association between SLR1 and OsMYC3.

Other comments.

Fig. 1, and associated narrative for BiFC assay (lines 141-146). The authors

negative control shows that no YFP signal is observed for the combination SLR1-cYFP/Gus-nYFP. They also should include the negative controls for SP8-nYFP, P2-nYFP, and M-nYFP.

Response: We agree with this comment and have added this data as Supplemental Fig. 1d.

Fig.1 and its legend, lines 746-756. Panels 1c and 1e have red stars next to specific bands. Explain the red stars in the figure legend. Also, should bands in 1d also be labeled with red stars? Be sure throughout the paper to include an explanation for the red stars adjacent to bands in the figures.

Response: Red asterisks next to specific bands have been added in Fig. 1d. We also clarified this in the figure legend, in Line 901.

Lines 150-153. The authors describe Co-IPs between SLR1 and SP8, P2, and M. In subsequent Co-IPs, the authors mention that the necessity of pre-treatment with MG132 to stabilize SLR1. Was MG132 also used in Fig. 1c-e?

Response: These Co-IP samples were not treated with MG132.

Lines 153-155. The authors conclude that SLR1 is a conserved interaction partner via its GRAS domain with the viral proteins SP8/P2/M. The evidence for interaction specifically with the GRAS domain only is observed through the yeast two-hybrid screen. For such a conclusive statement, it would be good if the interaction with the GRAS domain was confirmed with either the BiFC assay or co-IP assay.

Response: Thanks, we agree with this comment and have generated individual SLR1^{DELLA} and SLR1^{GRAS} mutants in an attempt to achieve specific SP8/P2/M-SLR1^{GRAS} binding using Co-IP assays. Overall, we obtained very similar results to the yeast two-hybrid screen that the different viral proteins SP8, P2 and M consistently interact with SLR1 through the conserved GRAS

domain. These data have been added as Supplemental Fig.1. See below Figure 8.

Figure 8. Mapping the domain of SLR1-interacting with different viral proteins.

Lines 313-317, and supplemental Fig. 5. Authors state, "... increasing the quantity of SLR1 progressively decreased the YFP fluorescence signals of the OsJAZ9-cYFP/OsMYC3-nYFP combinations." Supplemental Fig. 5 presents only a single agroinfiltration of SLR1. In this type of assay, it is important to provide the OD600 for the agrobacterium isolates used for agroinfiltration. Finally, to evaluate the effect of SLR1 on OsJAZ9-cYFP/OsMYC3-nYFP, the two treatments need to be conducted on half leaves of the same *N. benthamiana* leaf, and multiple leaves should be evaluated in this manner.

Response: We have analyzed the fluorescent intensity by Image J and have provided more detailed descriptions in the Figure legend and Methods (Lines

547-553).

Lines 325-327. Supplemental Fig. 7 shows results of a Y2H screen for evidence of interaction of GRAS domain with OsJaz proteins. As with Fig. 1, are Y2H interactions always representative of the interactions in plants?

Response: We have now confirmed OsJAZ9-SLR1^{GRAS} by Co-IP assay (Supplemental Fig. 11b). See below Figure 9.

Figure 9. Mapping the domain of SLR1-interacting with OsJAZ9 protein.

Reviewer #3 (Remarks to the Author):

In this study entitled “Independently evolved viral effectors convergently suppress DELLA protein SLR1-mediated broad-spectrum antiviral immunity in rice”, the authors found three rice viruses encoded proteins (SRBSDV SP8, RSV P2 and RSMV M) interacted with the general target protein SLR1 in vitro and in vivo, and trigger rapid degradation of SLR1 by promoting the interaction of GA receptor OsGID1 with SLR1, and also diminish the ability of SLR1 to activate JA signaling by disassociating SLR1 from the OsJAZ-OsMYC2/3 complex, leading to repressing SLR1-mediated broad-spectrum antiviral defense to viral infection. The findings are of significance to the pathogen-host

interaction field and related fields. This is an original work, the data support the conclusions and claims, and the methodology is sound and the work meet the expected standards in the field. Therefore, the work is suitable for publication in the journal after minor revisions.

Response: We appreciate the positive comments made by the reviewers #3.

My comments on minor revisions:

1.English language should by extensively checked, pay more attention to verb tense agreement.

Response: This has been checked.

2.Line383, please delete one "that".

Response: Done.

3.Most references cited are not unified in format, in particular authors' name.

Response: Thanks for the reviewer's advice. We have corrected these mistakes carefully in our revised manuscript.

Reviewer #4 (Remarks to the Author):

In this study, the authors reported that the molecular mechanism of infection by viruses is related to GA signaling. First, inspired by previous studies showing that viral proteins function as key factors in JA and auxin signaling, the authors found that SLR1, a key factor in GA signaling, binds to viral proteins and promotes degradation of SLR1. Using rice SLR1 knockdown mutants and overexpressors, as well as rice plants overexpressing viral proteins, they showed that SLR1 functions to suppress viral infection and that its action is inhibited by viral proteins. In addition, based on their previous results, the authors investigated the effects of SLR1 and viral proteins on the action of JA signaling key factors JAZs and MYC2/3 using molecular biological and

physiological studies to clarify the regulatory mechanism of GA-JA signaling in rice. Finally, they concluded that viral proteins hijack this mechanism to spread the infection.

The authors succeeded to show that the viral proteins bind to and regulates the degradation of SLR1, which could be the first direct observations of a molecular mechanism for direct crosstalk between pathogen infection and GA signaling. Thus, I consider that the findings could be potentially evaluated as novel in terms of plant molecular physiology and pathology. However, some of results presented here differ from previously reported observations, particularly on GA signaling, and they need to carefully check the reliability of these results. Some of the problems are pointed out below.

Fig. 2e shows that overexpression of SP8 or P2 greatly reduced the amount of SLR1 in the plant. Despite this, the height of these plants in panel f/h was similar to that of the control plants (Nip), which is completely different from the previous observation. The authors also noticed such discrepancy and gave the following excuse; No differences in plant height were observed between these plants, perhaps partly because there are only small amounts of endogenous SLR1 in Nip background rice plants, and although SLR1 was degraded in the transgenic plants, the differences were insufficient to cause phenotypic effects. However, this explanation is not consistent with the results of Fig. 2e. In fact, SLR1 in the Nip background is clearly observed in Fig. 2e. In addition, overexpression of SP8 or P2 greatly reduces SLR1 in plants, so the differences were not insufficient. In this paper, I often see this kind of interpretation that favors the authors' hypothesis (in one case, the GA signal is de-repressed as a result of the degradation of SLR1, and in another case, SLR1 is degraded but the GA signal remains unchanged). Such an attitude undermines the credibility of this paper.

Response: Thanks, we really with the agree reviewer's comment and have deleted the inappropriate description in the revised manuscript. We found it

was interesting that *SP8-ox* and *P2-ox* plant lines were morphologically distinguishable from wild-type *Nip* plants under GA₃ treatment (Fig. 2e-h) while *in vitro* protein degradation assays showed that SP8/P2 notably attenuated endogenous SLR1 accumulation in the presence of GA₃, indicating that these viral proteins elongate the second leaf sheaths of rice seedlings in a GA₃-dependent manner. Most importantly, the observed effects on the elongation of rice seedlings were remarkably increased in hybrid rice strains, in which *SLR1-GFP/SP8-ox* and *SLR1-GFP/P2-ox* plants were taller and had more panicles than *SLR1-GFP/Nip* controls (Supplemental Fig. 7), further supporting our claim that overexpression of SP8 or P2 greatly reduced the amount of SLR1, thereby leading to obvious elongation of rice seedlings.

As for the concern that no obvious height difference was observed between *SP8-ox* or *P2-ox* and *Nip* plants under normal growth conditions, we speculate that plants maintain a dynamic balance during the whole growth and developmental process. In the abovementioned transgenic rice plants, constitutive expression of SP8 or P2 in *planta*, acts as an elicitor triggering rapid degradation of SLR1, a key repressor of GA signaling, which probably in turn forms a negative feedback regulatory module to fine-tune plant growth, thus allowing adaptive survival against internal metabolic challenges. However, addition exogenous GA₃ treatment or constitutive overexpression of *SLR1* might be able to break down this dynamic balance of plant fitness *in vivo*, so that *SP8-ox* or *P2-ox* transgenic plants are morphologically indistinguishable (see below Figure 10).

Figure 10. Viral proteins SP8 and P2 manipulate SLR1 in GA-dependent manner.

Fig.2c and 2d also have a major problem. In this experiment, the authors used "total protein extracted from seedlings" to observe the degradation of SLR1 by GA₃, but the degradation of SLR1 is not reproducible. In fact, its degradation pattern of "+His+GA₃" and "+GST+GA₃" is not the same, but the results of both are clearly different. Furthermore, in "+His+SP8+GA₃", SLR1 seems to be degraded very slowly, while in "+GST+P2+GA₃", SLR1 is partially degraded in the first 60 minutes and no further degradation is observed. Based

on these results, they should assume that they cannot reproducibly observe the serial change of SLR1 by GA in this experimental system.

Response: We thank the reviewer #4 for pointing out this relevant issue which has enabled our manuscript to be more rigorous. We have repeated the *in vitro* cell-free protein degradation assays using unified His-SP8 and His-P2 vs. the control His-TF. Our results provide strong evidence that the degradation of endogenous SLR1 was remarkably accelerated in the presence of SP8 or P2 protein compared with the control His-TF (Fig. 2c and d). Please see our response to this reviewer's comment #2.

Gibberellin modulates multiple aspects of plant behavior, it was generally perceived by the nuclear receptor GID1, which then interacts with the DELLA nuclear proteins and promote their degradation. However, our results also observed that these distinct viral proteins (SP8, P2 and M) directly interact with individual SLR1 and OsGID1 in the absence of GA₃ (Fig.3c-d and Fig.3g-h), while the addition of GA₃ induces the functional interaction between SLR1 and OsGID1. Increasing the amounts of either TF-His-SP8 or MBP-His-P2 increases this interaction (Fig.3g and 3h), thereby facilitating the formation of the functional complex SLR1-SP8/P2-OsGID1. All of these point to the conclusion that SP8 and P2 proteins act as scaffolds to bring SLR1 physically adjacent to its receptor OsGID1, while making GA₃ recognition indispensable to the functional degradation of SLR1.

The results in Fig. 2g and 2i are also questionable. This experiment was conducted to investigate the effect of GA₃ concentration on the elongation of rice seedling. In a previous experiment, the GA₃ concentration on the elongation of the second leaf sheath length of rice seedling was reported to be 10⁻¹³ to 10⁻¹⁰ M (e.g. Planta 205(2):145, 1998). Since the GA concentration used in this experiment is much higher than that, they cannot discuss effect of GA concentration. Furthermore, the fact that the leaf sheath length of SP8-ox and P2-ox is longer than that of Nip even at very high concentrations of GA

(even though GA signaling is fully saturated) suggests that this difference is due to something other than GA signaling.

Response: Thanks for your valuable comments, and we really appreciate the reviewer for pointing out this relevant issue that we had previously overlooked. In our revised manuscript, as the reviewer suggested, we have added different concentrations of GA₃ (0.1 μM, 1 μM, 2 μM, 5 μM and 10 μM) treatment to investigate the effect of GA₃ concentration on the elongation of *SP8-ox* and *P2-ox* transgenic plants. Upon statistical analysis, we realized the interesting fact that in the presence of low concentrations of GA₃ (0.1 μM and 1 μM), the second leaf sheath lengths of *SP8-ox* and *P2-ox* were nearly indistinguishable from wild-type *Nip* seedlings, whereas there were differential sensitivities at higher concentrations (2 μM, 5 μM and 10 μM) (Fig. 2e-h), suggesting that *SP8-ox* and *P2-ox* transgenic plants enhanced degradation of endogenous SLR1, making them more sensitive to elongation of the second leaf sheath in a GA₃-dependent manner. See below Figure 11.

The reviewer notes that earlier research showed that the most effective concentrations for promotion of leaf sheath growth by GA₃ (10⁻¹³ to 10⁻¹⁰ M) were greatly exceeded (saturated) by our use of 10⁻⁷ to 10⁻⁵ M GA₃. However, these experiments used very different methods of analysis. In the paper cited, “gibberellic acid was applied to the coleoptile of each seedling at the first-leaf stage, when the tip of the blade of the second leaf had just emerged” (Planta 205:145, 1998), whereas we used rice seedlings planted into nutrient solutions containing different concentrations of GA₃ for about 7 days, a procedure adopted in many important studies involving hormone treatment (De Vleeschauwer et al., 2016, Lan et al., 2020 and Yang et al., 2012).

Figure 11. GA_3 sensitivity of plants constitutively expressing SP8 and P2 proteins.

References:

1. Matsukura C, Itoh S-i, Nemoto K, Tanimoto E, Yamaguchi J. Promotion of leaf sheath growth by gibberellic acid in a dwarf mutant of rice. *Planta* **205**, (1998).
2. De Vleeschauwer D, et al. The DELLA Protein SLR1 Integrates and Amplifies Salicylic Acid- and Jasmonic Acid-Dependent Innate Immunity in Rice. *Plant Physiology* **170**, (2016).
3. Yang D-L, et al. Plant hormone jasmonate prioritizes defense over growth by interfering with gibberellin signaling cascade. *Proc Nat Acad Sci USA* **109**, (2012).
4. Lan J, et al. Small grain and semi-dwarf 3, a WRKY transcription factor, negatively regulates plant height and grain size by stabilizing SLR1 expression in rice. *Plant Molecular Biology* **104**, (2020).

In Fig. 3, the authors examine the formation of GID1-SLR1-SP8/P2 in transiently infiltrated tobacco leaves. They showed that the SP8/P2 protein binds to GID1 and SLR1 individually (presumably in the absence of GA), but they did not say whether GA is required for this GID1-SLR1-SP8/P2 formation. What effect the involvement of SP8/P2 in OsGID1-SLR1 interaction has on the

degradation of SLR1 (e.g., responsiveness to GA and rate of SLR1 degradation) is very important for this paper. However, this paper does not mention this at all. This should perform some new experiments using in vivo system (not in vitro) to answer these questions.

Response: Thanks for the reviewer's helpful suggestion. We agree with this comment and have now added in *in vitro* pull-down assays in the revised manuscript to make it clear whether GA₃ is required for the formation of OsGID1-SLR1-SP8/P2. The new results support our previous conclusion that viral proteins SP8/P2 interact separately with SLR1 or OsGID1 in a GA₃-independent manner (Fig. 3c, d and Supplemental Fig. 4), but that SP8/P2 promote the association of SLR1 and OsGID1 in a GA₃-dependent manner (Fig. 3g, h). See below Figure 12.

Figure 12. Interactions among viral proteins with SLR1 and OsGID1.

Fig. 4 also contains a similar problem. Why is Ri-SLR1 not growing taller? Is

the amount of SLR1 really changing under these conditions? If so, does the change in the amount of SLR1 affect GA signaling? As expected, the SLR1-GFP plants show dwarfism. Then, how much accumulation of SLR1 occurs in these plants to suppress its plant height? In Fig. 4, the authors observed only changes in viral proteins and virus symptoms, but paid no attention to the quantitative changes in SLR1 even though they want to discuss the relationship between SLR1 and viral propagation.

Response: We are grateful for the reviewer's helpful comments. During the whole period of phenotypic analysis of SLR1-related transgenic rice plants, we consistently recorded the significant dwarfism of *SLR1-GFP* and the taller stature of *RNAi-SLR1*, which is consistent with a previous study (Liao et al., 2019). In Fig. 4a, the rice seedlings were initially cultured in an artificial greenhouse, which might be suboptimal in the late growing period. Upon RSV infection, we mainly focused on the areas of typical yellow stripes and curling or death on the young leaves to evaluate viral propagation. Importantly, in Fig. 5a, where plants were naturally grown in the field, *RNAi-SLR1* was obviously taller than the *LS* control.

References:

1. Liao Z, et al. SLR1 inhibits MOC1 degradation to coordinate tiller number and plant height in rice. *Nature Communications* **10**, (2019).

About the result of Fig. 7b, they discussed that "endogenous SLR1 was more rapidly degraded". Of course, they cannot make such discussion on the kinetics of SLR1 degradation by using this result. The discussion on the kinetics of SLR1 degradation and that of GA-dose dependence should be very important for this paper, because when the virus components are really involved in the GA signaling via SLR1, the events caused by viral proteins should mimic the stereotype GA-response. The authors should pay attention to the plant physiological aspects of GA-induced phenomena and re-examine the

physiological phenomena during viral infection, which is the subject in this paper.

Response: we thank the reviewer for pointing this out. In response to the concern of the reviewer, we have provided new kinetics results of SLR1 degradation in our revised manuscript (Supplementary Fig. 12). See below Figure 13.

Figure 13. M protein provoked rapid degradation of SLR1 in cell-free system.

The presentation and explanation of Fig. 8 is inappropriate and is not a good summary of this paper. When SLR1 binds to OsMYC2/3, it facilitates JA signaling resulting in inhibiting the rice growth/development. This figure does not show that state. On the other hand, in the presence of viral infection (viral proteins), SLR1 degradation is accelerated and binding to OsMYC2/3 is inhibited, resulting in accelerated binding of OsJAZ and viral proteins to OsMYC2/3 (right side of viral infection in the figure). In this state, JA signaling should be reduced, and plant growth should be free from its inhibitory state and should be growth-promoting. However, in the figure, this state is represented by a dwarfed plant with a T-formed line indicating inhibition. This dwarfed plant (probably) indicates a state in which viral resistance is weakened due to reduced JA signaling and plant growth is inhibited as a result of increased viral infection, but it is clearly inappropriate to represent this as a T-formed line on the plant.

Response: Thanks. We have modified the model in Figure 8 as suggested. Considering the complicated functions and mechanisms of phytohormones, we speculate that plants might have a dynamic balance to adapt their fitness. In normal conditions, part of SLR1 binds to OsMYC2/3 or OsJAZ proteins, activating a branch of JA signaling resulting in inhibition of rice growth/development, while the remainder of OsJAZ proteins continue suppressing OsMYC2/3-mediated JA activation, thereby achieving a balance to help plants survive in moderate growth conditions. This regulatory physiological reaction is purposely labeled with double arrows in the working model.

We have also wondered whether a virus might encode several multifunctional virulence factors that can target different pathways to subvert host defense. If that is the case then symptoms of viral infection cannot be attributed to any specific effector alone, because the other factors may also assist in promoting virulence. More importantly, when plants encounter adverse environmental factors, they generally activate a “moderate” defense response at the expense of normal growth to reallocate their limited resources to ensure survival and such modulation of the growth–defense balance partly results in stunted growth.

We really thank the reviewer for raising this interesting issue. Although a meaningful discussion of this important but complex issue is out of the scope of this study, we are presently preparing a separate manuscript to discuss in more detail the role of the identified viral proteins in disturbing the SLR1-mediated broad-spectrum antiviral network.

Minor comments

About Fig. S5, the text mentioned that they quantified the YFP signals by OsJAZ9-OsMYC3 complex under the presence/absence of SLR1. However, there is no information on how to quantify the signals. They need to describe the details about this experiment.

Response: We have provided more details in Methods (Lines 547-553).

Reviewer #1 (Remarks to the Author):

The authors have alleviated all my concerns and the manuscript is considering me ready for publication. There is only one minor correction that is required. In the rebuttal letter the authors cite the papers in which the qPCR assays were developed. They should incorporate these references into Supplemental table 1, so that researchers that would like to follow up on their experiments know where to look for the information about the assays.

Reviewer #3 (Remarks to the Author):

All the comments I raised have been addressed, therefore, the revised manuscript is suitable for publication in the journal.

Response to Reviewers' Comments

We are very pleased to receive the reviewers' positive decisions about our paper. We have added the detailed information as suggested by the reviewers. Comments received are shown in black below, with our response in red font.

Reviewer #1 (Remarks to the Author):

The authors have alleviated all my concerns and the manuscript is considering me ready for publication. There is only one minor correction that is required. In the rebuttal letter the authors cite the papers in which the qPCR assays were developed. They should incorporate these references into Supplemental table 1, so that researchers that would like to follow up on their experiments know where to look for the information about the assays.

Response: We thank the Reviewer#1 for the positive comments. We agree with the reviewer and we have incorporated the related references into the revised Supplementary table 1.

Reviewer #3 (Remarks to the Author):

All the comments I raised have been addressed, therefore, the revised manuscript is suitable for publication in the journal.

Response: We thank the Reviewer#3 for the positive comments.